

# Trapnell's Upper Valley Soils of Zambia: the production of an integrated understanding of geomorphology, pedology, ecology and land use

Nalumino L. Namwanyi[1], Maurice J. Hutton[2], Ikabongo Mukumbuta[1], Lydia M. Chabala[1], Clarence Chongo[1], Stalin Sichinga[3], and R.Murray Lark[4]

[1]University of Zambia, Great East Road Campus, Lusaka, Zambia
[2]University of Nottingham, University Park Campus, Nottingham, NG7 2RD, United Kingdom
[3]Soil Survey Unit, Zambia Agriculture Research Institute, Mt Makulu Central Research Station, Chilanga, Lusaka, Zambia
[4]University of Nottingham, Sutton Bonington Campus, Loughborough LE12 5RD, United Kingdom

**Correspondence:** R.M. Lark (murray.lark@nottingham.ac.uk)

**Abstract.** The Ecological Survey of Northern Rhodesia, undertaken in the 1930s under the leadership of Colin Trapnell, was a seminal exercise to relate soil, vegetation and agricultural practices through intensive field observation. In this article we examine early activities of the survey in the Upper Valley region around the Kafue Flats and the neighbouring plateau where Trapnell recognized how geomorphological processes of normal erosion gave
rise to distinctive soils with associated vegetation communities and considerable potential for crop production. We consider how Trapnell's approach to field work gave him a particular insight into how soil conditions constrained agriculture in the Zambian environment, the adaptive value of traditional practices, and how these were developed as communities moved and responded to social, economic and environmental change. We argue that Trapnell's work was innovative, and that distinctions must be drawn between his understanding and what has
been called the ecological theory of development. Close attention to Trapnell's experience could inform modern efforts to understand indigenous knowledge of African soils and their agricultural potential.

## 1   Introduction

The pursuit of food security in sub-Saharan Africa requires understanding of soil resources. Two sources of understanding, superficially rather different, are indigenous or traditional soil knowledge, and legacy soil information
from surveys undertaken in colonial or early post-colonial periods. In this article we examine one such legacy survey and the information it provides, including what was recorded about traditional agricultural practices.

The Ecological Survey of Northern Rhodesia (now Zambia), which C.G. Trapnell led through the 1930s, (Trapnell et al., 1947), has been studied as an example of colonial science (e.g. Tilley, 2011) and used as a resource for studies on agricultural, environmental and social change (e.g. Moore and Vaughan, 1994). In the immediate aftermath
of the survey its methodology and findings were applied in locally-focused surveys in the country (Allan et al.,





1948; Allan, 1949); and it subsequently provided the basis for the Zambian contribution to the Soil Map of Africa (D'Hoore, 1964). The significance and novelty of Trapnell's work, using an ecological model to support inference about soils and agricultural practices in extensive survey, has been recognized (e.g. Young, 2017), and the wider significance of an ecological survey as a basis for a certain understanding of colonial development has been ex-

25 plored by Bowman (2011) and Speek (2014). However, Trapnell's work has been treated as something of a curiosity in the history of soil science, a side-note to the account of pedology in British Tropical Africa which is focused on the East African Soil Map (Milne, 1936). A review of soil survey in Africa by Dalal-Clayton (1988), while recognizing the pioneering ecological structure of Trapnell's map units, does not attribute any originality to his treatment of soil and landform.

Our contention is that Trapnell's work is very relevant to questions about the soil and food security in contemporary Africa. However, we also maintain that the evaluation of such sources requires cross-disciplinary collaboration between natural scientists and historians to evaluate the original surveys in their context. This article offers a reading of Trapnell's traverse notes as the record of production of integrated knowledge of soils, vegetation and land use by a team comprising scientists (Mukumbuta, Chabala, Sichinga, Lark) and historians (Namwanyi, Hut-

ton, Chongo). We evaluate Trapnell's work critically, considering how his field methods, in their colonial context, shape his findings, and their value and lasting significance. We also examine the distinctive methods of ecological survey, considering soils in their geormorphological setting, and the expression of their properties in the vegetation and the new and traditional agricultural practices they supported (or failed to support). The early work in the Upper Valley is particularly instructive for this because the role of geomorphological processes in controlling

the spatial pattern of soil variation, and so the capability of land, was particularly clear. At the same time economic factors (notably the development of the Livingstone to Lusaka railway line which opened up new markets) and the politics of colonialism with expropriation of land from African farmers for commercial use by Europeans, were major drivers of rapid change, the sustainability of which was moot.

We undertook close reading of Trapnell's Traverse Records from the Upper Valley as published by Smith and

45 Trapnell (2001) in Volume 1 (1932 – 1934), along with Trapnell's correspondence held in the archive at Royal Botanic Garden, Kew, London. In addition to these, we examined reports of the Northern Rhodesian Department of Agriculture held in the National Archives of Zambia (NAZ), Ridgeway, Lusaka, concerning the Ecological Survey and its activities. Published materials from the survey, and other unpublished syntheses, were also examined, including the two Ecological Survey reports, Trapnell and Clothier (1937), Trapnell (1943) and the final Soil-Vegetation

map. We also examined the proceedings of the 1932 and 1934 meetings of East African soil scientists as context for Trapnell's work and the first published account of his findings and methods (Milne, 1932, 1935).

In section 2 of this paper we give an overview of geomorphology in Trapnell's account. Section 3 summarizes the colonial perspective on African farming practices and the commissioning of the Ecological Survey. In section 4 we present our account of Trapnell's activities in the Upper Valley, based on close reading of the field records



and other unpublished reports. Sections 5 and 6 examine the presentation of the Upper Valley environment in, respectively, early syntheses of the Ecological Survey's outputs and its published reports.

## 2   Overview: Geomorphology in Trapnell's classification

Webster (1960), writing on the basis of field experience of soil survey in late-colonial Northern Rhodesia in the 1950s, suggests that climatic soil zones did not become a dominant model in Africa because of the widespread
influence of geomorphological processes and the age of the land surface on soil distribution there. He notes that, in the Zambian setting, uplift since the Karroo, peneplanation and faulting and the consequent variations in *age of the land surface or the alterations which have taken place in its relief* (Trapnell and Clothier, 1937, quoted by Webster) are key to understanding soil variation. Webster (1960) notes that these geomorphological influences on soil properties were recognized by Trapnell and Clothier (1937), and cites the use of Plateau and Upper Valley as
topographical terms to denote contrasting soil environments. In an overview of the soils of Zambia, Webster (1960) refers to the Upper Valley where normal erosion of the old Plateau surface leaves residual or colluvial soil material with a reserve of weatherable minerals, contrasting with the deeply weathered Plateau surface. It is this physical process which underlies the ecological differences on which Trapnell first distinguished the Upper Valley unit from the surrounding Plateau, and which also accounts for the fertility of the Upper Valley soils, their importance
in traditional agricultural systems and their significance for agriculture in Zambia when Trapnell was doing his field work.

Trapnell was primarily an ecologist and his approach was strongly influenced by the vegetation survey completed by Henkel (1931) in what was then Southern Rhodesia. The Upper Valley environment was recognized initially on the basis of its distinctive vegetation (Trapnell and Clothier, 1937), but Trapnell identified the importance
of erosional processes from the onset, see Trapnell (1935). Cole (1963) states that Trapnell was *concerned primarily with the coincidence of physiographic types and climate regimes* but this does not bear examination. The first published output from the Ecological Survey (Trapnell and Clothier, 1937) introduces the Upper Valley on the basis of its geomorphological origin and links this explicitly to its ecological and agricultural significance. Further, Trapnell and Clothier (1937) observe that, in a region like the Upper Valley undergoing normal erosion, soil formation
was takes place under a climate regime very different to that in which the genesis of Plateau Soils was initiated.

Little was known about the geomorphology of Zambia at the time of Trapnell's field work. The first synthesis of the geomorphology of Northern Rhodesia was by Dixey (1944). Earlier work focused on the Copperbelt in the north-west of the country. The first edition of Lester King's *South African Scenery*, (King, 1942), which gives a synoptic account of the regional landscape was not published until Trapnell's field work was complete. Topographic
mapping outside areas of particular economic importance, mainly the Copperbelt, was sparse at the time of Trapnell's field work (Haines, 2015). The first map with 500-foot contours was published in 1939 (Dixey, 1944) so, at least during the period of field work we examined, Trapnell did not have access to this information. His interpretation of



geomorphological processes was therefore limited to what he could see on foot and from limited airphoto cover, which was not stereoscopic.

## 3   Overview: 'Native agriculture', contrasting views, and the genesis of the Ecological Survey

'*The acquaintance of the Ba-ila with the principles of agriculture is very slight ; of fallowing, rotation of crops, manuring, seed selection, they know nothing. … Their present methods are extremely wasteful, both of labour and land.* Smith and Dale (1920)

This colonial assessment of African agriculture, the first author was a missionary, had been challenged before
Trapnell's field work, (Tilley, 2011). Homer Shantz, from the U.S. Department of Agriculture (see section 4.2) participated in the 1923 – 1924 African Education Commission tour of eastern and central Africa, and reported his findings, (Shantz, 1925). He wrote *The agricultural methods of the Natives in Africa have often been condemned as shiftless, wasteful and destined to decrease the productivity of the country … but there are many testimonies in the literature to the effect that the Native is an excellent agriculturist.* He went on to note that practices such as shifting
cultivation were routinely condemned, but argued that they were adaptive, and more effective at the restoration of fertility and soil physical quality than any alternative. He pointed to the effectiveness of African soil selection methods for matching crops to sites.

Shantz recognized that, at the time of writing, there was a widespread shift of focus among colonial administrators from European to African agriculture, and that many had a genuine interest in understanding traditional prac-
tices. Tilley (2011) notes that Shantz's views were regarded sympathetically by some British scientists and politicians, including the Undersecretary for the Colonies, William Ormsby-Gore. She highlights the work of Faulkner, Director of the Department of Agriculture in Nigeria from 1922, who prioritized the study of African farming.

In 1924 the administration of Northern Rhodesia was transferred from the British South Africa Company to the Colonial Office. The overall record of European settler farmers in the colony was not good. While there had been
short-lived successes with some crops (such as cotton), sustained production has not been achieved, which T. McEwen, the colony's chief agricultural research officer attributed to a lack of knowledge of plant ecology in the Zambian environment (Speek, 2014). The acting director of agriculture in the new colony, John Smith, reflecting the changing focus of attention to African farming, initiated two linked research projects in 1927: field experiments on African shifting cultivation methods led by Unwin Moffat, and a programme of ethno-agrobotany
undertaken by T.C. Moore whose team collected seeds and information on agronomic practices from across the country (Tilley, 2011).

In this context of new thinking about agriculture and its role in the future of the colony that the plans for the Ecological Survey were developed (1927 – 1928). Key to this were recommendations from R. Bourne, of the Imperial Forestry Institute at Oxford University, for a multi-disciplinary team of geologists, foresters and agriculturalists
with good knowledge of the local flora, to undertake an ecological survey aided by air photography. There was gen-





eral support in Northern Rhodesia for the survey, but this concealed divergent understandings of its purpose and the general shape of the policy it would enable (Speek, 2014). Bourne was of the view that European cultivation should be discouraged, and that Africans would be the main agents in development of land resources. Smith, in contrast, wanted research to support sound subsistence farming by Africans, but not competition with Europeans

in commercial production. According to Baldwin (1966), a widespread view among Northern Rhodesian officials was that food supply for the mines could be sustained by domestic production, only if this was undertaken by European farmers, requiring further immigration. This was one reason for the policy of resettling African communities who lived on productive land close to the railway. This particular conflict over agricultural policy paralleled broader unease within the settler community provoked by the Passfield Memorandum, asserting 'native

paramountcy' as a key principle of Britain's colonial policy (Wetherell, 1979). The NR legislative council responded that '*the British Empire is primarily concerned with the furtherance of the interests of British subjects of British race and only thereafter with other British subjects, protected races [etc.]*' (Colonial Office, 1930).

In this context, Smith forwarded the proposal for the Ecological Survey to the Empire Marketing Board (EMB) with the suggestion that it would support improved livestock production by Europeans. Nonetheless, as Speek

(2014) notes, the Governor of the Colony undertook that further settlement schemes for European farmers would wait on the results of the survey.

The bid for EMB support was unsuccessful, but the proposal was developed with technical input from the Royal Botanic Gardens at Kew, which emphasized both the potential to identify land for export crops, and to base development in a young colony on a scientific survey undertaken before the widespread impact of settlement or change

in African farming methods (Speek, 2014). H.C. Sampson, economic botanist at Kew, stated that the survey should entail '*enquiry into indigenous agricultural practice such as crops, varieties, soils, seasons, and their association one with the other and with the natural vegetation.*' (Sampson, 1928). Funding was provided by the Colonial Development Fund, but it was not until 1931 that Trapnell was appointed as an ecologist to lead the work, and the survey was eventually inaugurated in 1932.

**4 Field work in the Upper Valley**

In this section we examine the survey activities which took place in the Upper Valley environment from 1932 – 1934. The primary source is the set of Trapnell's field traverse records, (Smith and Trapnell, 2001), but we also refer to reports from the Ecological Survey contributed to Department of Agriculture Annual reports, and to other reports by Trapnell and Clothier available in the National Archive of Zambia.

We outline the itinerary of field work in the Upper Valley area. We then review the information available to characterize Trapnell's field survey methods, and highlight some aspects of these that emerge from a close reading of the field records for the Upper Valley. We then discuss what these records show Trapnell to have observed in the Upper Valley and associated Plateau, and how this contributed to the emergence of the model of the Upper Val-



ley as a distinctive environment created by geomorphological processes, developing a distinctive vegetation, and
consequently with distinctive potential and challenges for agricultural use in Trapnell's time.

### 4.1   Reading the traverse records

Trapnell's traverse observations recorded in field-notebooks in Zambia between 1932 and 1943 were transcribed,
by P. Smith with Trapnell's assistance, and published by the Royal Botanic Gardens, Kew (Smith and Trapnell, 2001).
The original notebooks are held in the Royal Botanic Garden's archive. In this study we used the records for the
survey activities listed in section 4.2. These comprised the traverses for June – July 1932, and August – October
1932 recorded as the 'Ila-Tonga traverses 1932' in Part 3 of Volume 1 'Western, southern and central Zambia' of
Smith and Trapnell (2001). Some sections of the traverses lying on the sedimentary land of the Kafue flats were
excluded. Some key sites and a generalization of the route based on coordinates of some recorded sites are shown
on Figure 3 with blue symbols for sites on the traverse other than major towns, and blue dotted lines generalizing
the route for the initial days of the Inaugural Survey in which Trapnell participated. Trapnell's and Clothier's visit
to land south of the Kafue River in August and September 1932 are represented in the same Figure, see sites with
purple symbols, and route generalized by the purple dotted line in Figure 3.

In addition, the records for the survey listed as 'Road Traverses, Southern and Central Provinces 1933–1934', also
in part 3 of Volume 1 were examined. These covered land south of the Kafue river (1933, Fig. 4), and both sides of
the river (1934, Fig 5).

At this stage in the study we undertook close readings of the traverse records listed above. By reference to lo-
cations listed in the traverses, and their coordinates where provided, we were generally able to situate the ob-
servations on the Soil and Vegetation Map (Trapnell et al., 1947), using a scanned and georeferenced version
(Mukumbuta et al., 2022b). The close reading of the records was done with two objectives. First, to examine Trap-
nell's general methodology, as it was developed during field work in the Upper Valley, and to identify limitations
which should be considered alongside his innovative approach to using vegetation cover as an integrating princi-
ple for information on soils and land use. Second, we identified locations at which information on soil conditions
and agricultural practices were recorded along with the vegetation. These observations were summarized in tab-
ular form and the Tables for the Ila-Tonga Traverses, Reserve IX (Sala) and the Road Traverses (1933, 1934) are
presented in the supplementary material (Tables S1 – S3).

Two further summaries of this material were produced. Table S4 in the Supplementary material puts together
observations of farming practice on Plateau or Upper Valley sites with notes on the vegetation, characterization of
the rotation practices and shifting cultivation, and any comments recorded on farming. Table S5 draws together
observations recorded by Trapnell about changes in farming practice which his informants told him about or
which he inferred from observation.



## 4.2 Field activities undertaken in the Upper Valley and associated Plateau environments

The Ecological Survey was inaugurated in June 1932 by R.S. Adamson from University of Cape Town. Adamson wrote a report on his visit, which included an itinerary, methods summary and a summary of findings discussed in more detail below (section 4.3). Trapnell (Trapnell, 1932a) provided a resumé of survey activities subsequent to

the inaugural survey in later 1932. Trapnell's traverse records (Smith and Trapnell, 2001) provide information on further road traverses in 1933 and 1934. These are the sources for the summary below.

The Inaugural Survey began in Mazabuka on June 13[th], and initially covered land north of the Kafue river. Adamson was accompanied by Trapnell and J.N. Clothier, Agricultural Officer to the Ecological Survey. After the first fortnight in the field Trapnell was taken ill, and his traverse records cease until August 1932.

The team continued to Kafue. In the second phase of the Inaugural Survey Adamson (13[th] to 23[rd] July, Adamson was accompanied by T.C. Moore and C.E. Duff, Agricultural and Forest Officers respectively. This second phase examined land south of Kafue. It is not clear whether, beyond Adamson's summary (Adamson, 1932), this second phase of field work contributed substantially to Ecological Survey outputs, and from August 30[th] to September 20[th] 1932 Trapnell and Clothier visited sites south of Mazabuka, and on the Kafue Flats, which covers much of the

same ground.

From October 6[th] 1932 Trapnell and Clothier visited the Mwembeshi Basin region, specifically to examine Sala Reserve (Reserve IX), see the region outlined by a solid red line in Figure 3. This study produced similar descriptions of soil, vegetation and agricultural practices to the Ecological Survey, but with observations concentrated in a smaller area.

The activity listed above, between April and October 1932, is described in the 'Ila-Tonga traverses' section in volume 1 of Smith and Trapnell (2001). That volume also contains records of road traverses in Southern and Central provinces from 1933 and 1934 (the precise dates are not always clear). These covered Plateau and Upper Valley environments north and south of the Kafue River.

## 4.3 Survey methods: External evidence

Formal methodological statements about the Ecological Survey and its practice are few. Perhaps the only contemporaneous account is a terse summary provided by Adamson (1932) regarding methods used in the inaugural traverses in his report to the Northern Rhodesian Government. According to Adamson (1932), the team travelled primarily by vehicle, noting vegetation along the route and recording it relative to the mileometer. At selected locations more detailed studies were made by foot or bicycle traverse. The ecologist(s) were concerned with soil and

indigenous vegetation, and the agricultural officer collected information on farming practices. Soil samples were collected from both cultivated and uncultivated soils, and sent on to central laboratories for analysis although, as noted by Trapnell and Clothier (1937), very little soil analysis was to be completed because of financial constraints.





Tilley (2011), reporting from an interview with Trapnell, gives some limited information on later practice by Trapnell and Clothier, which contrasts with the description given by Adamson (1932). Travel was primarily on
foot, with the assistance of a team of porters and one or more translators. In a village they would meet with elders and ask questions described as 'routine' about key practices – land selection, clearing, planting, the duration of cultivation and extent of rest periods. However, Smith and Trapnell (2001) state that the settled practice of the Ecological Survey emerged in the course of the survey of Barotseland undertaken from May to August 1933, so the procedure was only emerging at the time of the first traverses in the Upper Valley, and the account given to Tilley.

Allan (1965) describes field survey procedures for land capability evaluation explicitly based on the Ecological Survey methods, in which Allan participated. But, as these included the survey of end-points of traverses, and use of prismatic compasses to mark them up, with clearance of ground to facilitate passage of the teams for more intensive survey of smaller areas than the Ecological Survey covered, it is clear that they tell us little about the original Ecological Survey itself.

Trapnell's (1937) article is ostensibly on the method of the ecological survey, but is rather a higher-level account of the hypotheses which early stages of the survey (principally of the Kafue Basin) were held to validate, so justifying later practice. Trapnell (1937) presents the Ecological Survey as a new kind of field study explicitly tied to two linked hypotheses; first, that vegetation type is directly correlated with the agricultural capability of land, and so with succesful farming practices on that land; second, that vegetation is correlated with soil type of soil properties,
and so with the agricultural capability of land. On this basis the Ecological Survey, primarily structured by the observation of vegetation classes, provides a basis to test this hypothesis. Trapnell treats the lower Kafue Basin stages of the Ecological Survey as a test of these hypotheses, the first one being validated because African cultivators who were interviewed by the surveyors recognized the same vegetation classes as the surveyors, used these classes in the selection of cultivation sites, employed different practices and crops on land under these classes, and, to vary-
ing extents, had a concept of vegetation type as an indicator of fertility. The second hypothesis could not be tested in terms of particular soil properties related to fertility, requiring laboratory analyses. However, Trapnell (1937) observes that the vegetation classes related to classes of the underlying soil, primarily defined with respect to physiography, and comparable with classes used in the East African Soil Map (Milne, 1936).

### 4.4   Survey methods: Air photography

Air photography began in Northern Rhodesia in a campaign (1927 – 1929) focused in the North West to facilitate the accurate mapping of mining concessions, planning of infrastructure to support the mines and, more speculatively, to aid mineral exploration. Air photography, and its conversion to topographic mapping, was undertaken by the Aircraft Operating Company (AOC), with the initial photography paid for by the Rhodesia Congo Border Concession Ltd and the cartography by the Colonial Office (Haines, 2015). Subsequently (1931) the North-
ern Rhodesian Government paid for some additional air photography in the Copperbelt and 15 miles either side of the railway line south of Mazabuka. This latter work was in collaboration with the Agricultural Survey Com-





mission to facilitate distribution of land to European farmers. The AOC, with a view to promoting the use of air photography in the colony, undertook an independent air survey of land in a block centred roughly on Lusaka and bordered at the south by the Kafue river. This photography was interpreted in terms of the vegetation types and

the agricultural potential of land by AOC's C.R. Robbins, who provided a report to the Agricultural Department. He described the use of parallel strips of vertical photography, supplemented by oblique views, for the delineation of different vegetation types by visual interpretation. Comments on the report, including specific comments from Trapnell, were forwarded to the Chief Secretary at Livingstone in June 1932 and January 1933 (Robbins, 1932). In the first set, Trapnell, who had not yet undertaken substantial fieldwork in the country, commented that the

photography clearly distinguished certain bush types, but not all (including the agriculturally important thorn country) but he was positive about the use of air photography as part of an overall survey procedure. Robbins's report was subsequently published (Robbins, 1934), and Trapnell's contribution to the description of some of the units is acknowledged. When the original report and the paper are compared it is apparent that Trapnell added more botanical and ecological detail, and also geomorphological information on colluvial and alluvial parent ma-

terial and dambos. Trapnell's comments, forwarded to the Chief Secretary at Livingstone on 17th June, just 4 days after the commencement of the inaugural traverse of the Ecological Survey, show that he had already familiar-ized himself not only with the vegetation of the plateau and Upper Valley but also, at least to some extent, with knowledge about the ecological potential of land under contrasting vegetation. There is no evidence that air pho-tographs were used in the field by Trapnell, but Trapnell and Clothier (1937) state that Robbins's photography was

used to produce the map that accompanies that report (Paragraph 68).

Robbins's report, with comments from Trapnell explaining his approach to survey, was shared with R. Bourne at the Imperial Forestry Institute in Oxford. Recall (section 3), that Bourne's enthusiasm for the potential of ecological survey supported by air photography has been an important factor in the initiation of the ecological survey. How-ever, Bourne was not impressed. His comments on the report (Robbins, 1932) made clear that he did not regard

the procedures described as adequate, in particular the lack of the substantial and interdisciplinary team which he had envisaged. He was sceptical about the general validity of the proposed connections between soil conditions, vegetation and agricultural potential on which the Ecological Survey was to be based. Although he acknowledged that the Ecological Survey activities were in an early stage of development he was prepared to make the statement *I cannot help thinking that Mr Trapnell's as well as Captain Robbins's investigations have not been sufficiently thor-*

*ough* (Minute forwarded from Imperial Forestry Institute in Oxford by the Director on 2ⁿᵈ Sept 1933, in Robbins (1932)). However, Bourne did state that if Trapnell had the evidence to back up claims about the 'mappable' re-gions then *his final report will be very valuable*. Trapnell was clearly concerned about this negative judgement. In a minute, (Trapnell, 1933), he stated the need to publish intermediate results to support his interpretation of African selection rules and their value for ecological survey. Lewin forwarded this minute to the Chief Secretary at

Livingstone with a covering letter which stated his strong support for what Trapnell was doing (Lewin, 1933). In this he stated '*I must confess that I read the final portion of Mr Bourne's memorandum with amazement. It is well*





*known that Mr Bourne holds very decided views on the subject of surveys of this nature. ...[O]ne cannot but feel that*
*he is somewhat prejudiced against a survey which is obtaining results by slightly different methods. ...The alternative*
*organisation suggested by Mr Bourne is probably the ideal but it would be expensive, cumbersome and in practice,*
*unless its personnel were of exceptionable calibre both technically and socially, might well fail where a survey or-*
*ganised on the lines of the one now in operation would succeed.*' He went on to state '*I have no hesitation in saying*
*that I consider the Ecological Survey to be the most important and usefuol activity which has yet been inaugurated*
*for the benefit of agriculture in Northern Rhodesia.*'

The Ecological Survey was not derailed by Bourne's negative judgement. However, in the light of the comments
of Trapnell (1933) it seems like that this criticism motivated the framing of the early phases of the ecological sur-
vey in terms of the testing of a hypothesis of the general ecological value of African land selection rules. This was
presented in a paper on the Ecological Survey's methods (Trapnell, 1937). It may also explain why Trapnell, in the
reports of the Ecological Survey, always emphasized the consistency of his topographically-defined soil classes
with those of the East African map, as independent validation of his method. It does, however underplay the orig-
300 inality of Trapnell's own work.

## 4.5 Survey methods: Evidence in the traverse records

In this study we explore the potential of an extended close reading of Trapnell's traverse records as a source for un-
derstanding his interpretation of one Zambian environment of particular interest for pedology and the emergence
of soil survey methods. Close reading of field records to elucidate the production of knowledge, the environment,
the role of field assistants and interlocutors and the realities of fieldwork practice has been undertaken in African
historical studies, e.g. Weintroub (2015) on Dorothea Bleek's ethnological and linguistic field work in the Kalahari,
Namibia, Angola and Tanzania. Here we focus on the routine methodology used in the Upper Valley, protocols (or
lack of them) for describing soil, vegetation and agricultural systems, Trapnell's sources and his approach to them,
and the particular focus of his interests.

We have relied primarily on the published version of Trapnell's records (Smith and Trapnell, 2001) for reasons of
accessibility (much of this work was done in Zambia, and during the COVID 19 pandemic when access to archives
was, at best, restricted). We have, however (RML) been able to make a direct comparison between some of Trap-
nell's notebooks in the Archive of the Royal Botanical Gardens at Kew (e.g. Trapnell, 1932b) and the publication.
Trapnell collaborated with Smith in the publication, including the transcription of the original notes, which are
315 not always very legible in the original. In places Trapnell's later comments on observations in the field notes are
included, but these are footnotes rather than interpolations so can be distinguished from the original material.
Where the published records and the originals do differ is that, in the latter, Trapnell used vernacular names to
refer to plant species, whereas in the published version these are usually rendered as the botanical names as used
in the published reports of the Ecological Survey (a table of synonymies with current botanical names is provided,
as well as lists of the vernacular names for plant species, ecological assemblages and cultivation systems). This is



helpful because the vernacular names for particular species change throughout the survey as Trapnell encountered speakers of different languages. The published records also include clear reproductions of Trapnell's field sketches, an important part of his methodology as we note below.

### 4.5.1   Daily practice of the survey

The traverse records provide a narrative of field work, indicating typically the start and end of each day's route, the time at which locations were reached, and the distance along the route at which changes in vegetation cover occurred. Locations may be settlements, often named for the headman. Trapnell often made sketches in his note book: maps, panoramic diagrams and topographical cross sections showing vegetation and land use in different slope positions. These are reproduced in Smith and Trapnell (2001). The cross-section sketches, now common-

place in textbooks of pedology to illustrate catenas or other soil-landscape patterns, put particular emphasis on general relationships between relief, soil conditions and landuse practices, reflecting an emerging model of how these were linked. The panoramas show specific local topography with arrangements of geology, drainage and vegetation, and are reminiscent of the oblique air photography produced by Robbins (1934) for interpretative rather than cartographic use.

Observations on vegetation, geology, soils, crops and agricultural practices are included in the records, often within the daily itinerary, but sometimes as a separate block of notes at the end. Comments are comprehensive and reflect wide ranging conversations with local informants. In addition to information about the crops in the ground, descriptions are given of the rotations, the shifts (how long cleared land was cultivated, how long land was fallowed), indicator species used for site selection, social observations (how much land a family cultivates, what land is cultivated for the chief), prices obtained for local products, foods resorted to in famine periods, and

changes in farming practice.

Adamson (1932) states explictly that ecologists and agricultural officers concentrated on their specialities recording soil and vegetation, and cropping practices respectively during the inaugural survey. This is reflected in Trapnell's traverse notes for the inaugural traverse, where observations on cultivation are relatively sparse. On 16[th]

June 1932 Trapnell comments that cultivation was proceeded by lopping and burning of trees, and speculated that shifts between cultivation sites were probably fairly frequent. However, there is no evidence that this information was provided by local informants. Otherwise, Trapnell's observations on farm practice in June 1932 were limited to noting where there was evidence of cultivation, past or present, and some observations of the crops under cultivation. From August 1932, however, the observations on farming become more frequent, and systematic,

with observations on rotation practices which clearly reflect engagement with informants. We do not have access to field notes made by J.N. Clothier or other agricultural officers engaged in the survey, but it is clear from the summary in Tables A1 – A4 that Trapnell soon began a more cross-disciplinary approach to his task then Adamson (1932) describes, and his observations in Reserve IX and on the road traverses provide the kind of information on agricultural systems and their setting under different vegetation types which was set out in Clothier's report on



African farming practices (Clothier, 1933). As noted in section 4.3 above, the settled field protocol emerged after the Barotseland survey (May – August 1933), but there is no apparent change in emphasis or approach from the initial Ila-Tonga traverses of 1932 to those recorded in and around the Upper Valley in 1933 and 1934, although these were road traverses and so somewhat atypical.

### 4.5.2  A Paucity of Protocols

Modern soil survey, vegetation survey or descriptions of agricultural systems use defined protocols to ensure that information is collected in a consistent, comprehensive manner. It becomes apparent on reading Trapnell's field notes that the Ecological Survey did not use formal protocols to record soil properties, vegetation or farming practices. While Adamson (1932) reports observed plant species using taxonomic names, Clothier (1933) reported on vegetation of the Kafue basin primarily with English vernacular names, whilst Trapnell in his original traverse

records typically referred to plants by the vernacular names used by his African informants, taxonomic names are substituted in Smith and Trapnell (2001). Trapnell's original practice of using vernacular names may reflect his dependence, at least in part, on informants for identification of species, and specimens were sent to Kew for identification (Trapnell, 1934b).

Similarly, there was no consistent way to record cropping practices. We are grateful to P. Smith for an email

exchange on the following examples (Smith, pers. comm.). In some cases a crop might be listed in a rotation sequence with an integer subscript, interpreted as the number of successive seasons in which it appears (e.g. at 'Chifusa's' on the Kalomo to Macha mission traverse leg in 1933 (page 531 of Smith and Trapnell (2001) volume 1, but this convention is not used everywhere. As Mukumbuta et al. (2022b) note, a hyphen and a solidus were each used to denote a rotation in some records and intercropping or mixed cropping elsewhere (or possibly alternative

crops at some point in the sequence), and at some locations (e.g. near Lwidi river, page 582 of volume 1) a cropping sequence is given as an enumerated list. The description of rotations and shifting cultivation practices by Clothier (1933) are generally easier to interpret than the accounts in Trapnell's traverse records, which may be a result of Clothier's formal education in agricultural science (Young, 2017).

At least as far as soil survey is concerned, the lack of standard protocols reflects the fact that procedures for field

description of soils were still emerging and unstandardized at the time, (see comments by Mukumbuta et al., 2022b). The first edition of 'The study of the soil in the field', (Clarke, 1936) was not published when the Ecological Survey began. Clarke's book provided the foundation for field manuals used subsequently in England & Wales and elsewhere. The 'Soil Survey Manual' of the U.S. Department of Agriculture was first published in 1937 (Soil Survey Staff, 1937).

Furthermore, as Trapnell explicitly recognizes in a later formal account of the survey methods (Trapnell, 1937), the brief and the resources available for the survey did not permit an approach based on a soil survey with extensive sampling or profile description. Indeed, due to retrenchments in spending within the Agriculture Department of Northern Rhodesia, and the loss of the Soil Chemist's post (Trapnell and Clothier, 1937), very few analytical data





on soils were available when the survey formally reported for the Central and Western regions (Trapnell and Cloth-
ier, 1937).

### 4.5.3  Sources of information

With whom did Trapnell speak? In the interview recorded by Tilley (2011) he states that 'elders' were his principal
informants, but Trapnell did not systematically record the source of his information in the traverse records. In the
traverse records villages are referred to by the name of the headman, e.g. 'Chonga's (31st August, 1932, page 381).
It is not clear here whether Chonga himself was the principal or sole informant. Trapnell was told, for example,
that shifts on this land (where agriculture was described as semi-permanent), would happen when a son moved
to a new area, which he would then cultivate until he died or chose to move on. It was not clear whether this
would apply to all 'sons' in the village, or just to the headman's. Among the Tonga, settlement might be patrilocal
or matrilocal, with a newly-married couple settling near either family, or near the man's mother, (Jaspan, 2017).
It is therefore likely that major shifts under Tonga semi-permanent agriculture might involve single cultivators
moving longer or shorter distances according to choice. Because Ila settlement was typically patrilocal (Jaspan,
2017) shifts might be over smaller distances. The records for this village, however, are not clear.

At Muchila's in the Upper Valley (18th September 1932, page 419 - 420), it is made explicit that Muchila was the
informant. *Muchila's people ... cultivate in* Afrormosia *bush. For cultivation he chooses by* Afrormosia *with* Acacia
campylacantha *mixed in.* It is not clear whether the information we are given about Muchila's soil selection reflects
a general practice of selecting land in *Afrormosia* bush, or perhaps a privilege for the headman to select a superior
class of land. Elsewhere there was evidence that such privileges were exercised. For example, at Mantanyani's (6th
March 1934, pages 566 – 567) where bush and dambo head cultivation was undertaken it was recorded that the
*Chief has a separate, large (dambo head)* Acacia woodii *garden on the best land.* Here Trapnell's record gives us
a picture of social stratification of land use practice. Elsewhere we cannot always be sure whether the records
refer to land use in just one social stratum, whether there is no such stratification or whether the description is a
generalized account of farming within which there might be some variation.

How did Trapnell speak with his informants? In the interview with Tilley (2011) referred to above it was stated
that he travelled with one or more interpreter. Interpreters are not named in the traverse records, so these indi-
viduals' roles and linguistic specialisms (and possible limitations in some settings, given the many languages to
which Trapnell refers) remain unknown. Trapnell certainly recorded names for crops, wild plants and ecological
units in a wide range of vernaculars. For example, the notes to volume 1 of Smith and Trapnell (2001) record 15
names for Sorghum (*Sorghum bicolor*) in 11 language groups, seven names for *Acacia campylacantha* used in eight
language groups and three words for 'dambo' in two language groups, including words distinguishing those with
or without streams. A total of 46 agricultural terms were recorded describing agricultural systems such as anthill
gardens, manured 'home' gardens near the village and various systems on dambos. The collection of this detailed
vocabulary clearly required considerable linguistic expertise on the part of the interpreter and a capacity to grasp





the importance of the nice distinctions based on ecological setting, drainage conditions and cultivation methods. However we do not know anything about the background of these individuals and the education or experience

which equipped them for the task.

At only one place in the traverse records read for this study do we find any attempt to transcribe the speech of an informant, Siabasuni, who describes the shrub *Phyllanthus engleri*, which has very toxic bark and roots, as *meninge skellem mouti* (page 380). This is Cikabanga, originating from Fanagalo, the *lingua franca* used in the mines of South Africa. We have here a glimpse of Trapnell and colleagues communicating with informants in a

430 pidgin, but the ethnobotanical and linguistic depth of the records as a whole clearly did not depend on this.

Trapnell used ethnolinguistic names in the traverse records to refer to territory, for example, on 3rd September 1932 (page 386) he notes of the survey's change of direction at a store by Nalubamba's *turn north into Ila country*. Similarly he notes in the record for a road traverse in 1933 (pages 529 – 530) that the survey was passing through *millet country* and comments that the *Matotela stop at Machili.* The significance of this comment is clarified in

part II of the report of the Ecological Survey for Northern and Western Rhodesia (Trapnell and Clothier, 1937), where (paragraph 95) he refers to the Matotela as *a backward people who cultivate bullrush millet on poorer up-land sands.* Ethnolinguistic groups provide a framework in the report to describe the variation of agricultural systems (paragraphs 89 – 97) and the map which shows the distribution of agricultural systems has 'tribal' names as well as the names used to describe particular systems of cultivation. The 'Tribal index' to volume 1 of Smith and

Trapnell (2001) contains 81 separate entries, not including recorded variant names. There are 13 entries in this index for the traverse descriptions examined in this study (excluding references to photograph captions or entries in itineraries). Table S6a shows these entries by group, and the associated language group. There are 28 entries in which the name is used to describe a village or some part of a village where the occurrence of two or more ethnic groups is noted. The next largest set (16) are plant names, with an additional seven records of wild or famine

foods eaten. Other entries refer to soil selection and cropping practices, and so transfer across to the description of agricultural systems by Trapnell and Clothier (1937). There are also entries on particular trading specialisms (the Bambala, for example, specialised in tobacco processing), and where groups might transfer cattle to graze under the oversight of another during certain seasons. There are also comparisons or comments made between ethnic groups by members of those groups or by Trapnell. For example, Trapnell met an Ila community which stated they

did not make gardens in *Phragmites* on the river banks … *"because they are true Ba Ila, not Batwa who have them"*, (page 411, CGT's quotation marks).

How far the tribal labels used by administrators, missionaries and others in colonial Africa reflected the self-understanding of the people themselves has been challenged in various studies (e.g., Ranger, 1989). The robustness of Trapnell's ethnolinguistic framework therefore requires examination. Posner (2003) notes that mission-

455 ary programmes to standardize languages for Bible translation, and later policy on languages for instruction in secular colonial schools, along with the homogenizing effect of mass movement of workers to mines lies behind the replacement of the linguistic diversity of precolonial Zambia with the contemporary situation in which four





languages dominate (Bemba, Lozi, Tonga and Nyanja). The emerging dominance of these four was recognized as early as the 1940s by administrators and anthropologists (Posner, 2003), although estimates based on obser-
460 vations from 1930 suggest that in the pre-colonial period there had been seventeen principal languages, among many more, with any one spoken by less than ten percent of the population.

The ethnolinguistic information in Trapnell's traverse records does not reflect this emerging homogenization. Table S6b shows a simplified form of the classification of nineteen languages or dialects in the Glottolog classification (Hammarström et al., 2023) which appear in Trapnell's 10 principal language groups. Trapnell's largest group
is IT (Ila-Tonga) with six principal languages or dialects. Four of these are in the Kafue subfamily of the Greater Eastern Botatwe group in the classification of Hammarström et al. (2023). The fifth remaining of the Kafue languages, Lenje, was treated by Trapnell as a separate language group. The Toka dialect in Trapnell's IT group is in the Toka-Leya-Dombe subfamily, closely allied with the Kafue languages. This comparison shows Trapnell paying close attention to the linguistic diversity of his informants, at least in so far as this provide what we would now call
ethnopedological or ethnobotanical information. Indeed, when he encountered distinct Ila and Tonga names for particular species he recorded these as such (e.g. the Ila name *Mukamba* and the Tonga name *Mupapa* for the Pod Mahogany tree *Afzelia quanzensis*, page 400).

### 4.5.4 The information which Trapnell collected

Trapnell's interlocutors provided a wide range of information relevant to his interests. In particular they described
plant species, tree or grass, used to select land for cultivation. Figure 7 shows a list of 28 distinct vegetation descriptions (species, genera, associations and one structural group 'tall grass'). All of these were recorded at least once as an indicator for soil selection (in one case, *B. flagristipulata*, a counter-indicator), either for general cropping (the 22 indicators and 1 counter-indicator with symbols on the plot) or for a particular crop – for example, *A. campylacantha* with *Setaria phragmatoides* was never given as a general indicator, but there are three cases where
it was named as an indicator of land for Sorghum. In some cases (three for the general indicators in Figure 7), an indicator is proposed for land which is second or third choice for cultivation (e.g. *Afrormosia spp*, and one out of 2 records for *Setaria sp*, indicated on the Figure by a light-green symbol).

Trapnell was also informed about plant species or general form of vegetation whose appearance in secondary succession indicated that fallowed land could be cultivated again. For example land might be cultivated again if
*Hyparrhenia* grass regrows, or if *the bush is high* (Nangoma's in the Upper Valley, page 542. Practices described to Trapnell included shifts, and rotations and the susceptibility of the local systems to drought, pests and damage by wild animals.

Trapnell also recorded wild species which were eaten, including those which were important in poor cropping seasons, which he referred to as 'famine foods'. For example, at Muchila's in the Upper Valley (18th September
1932, page 419 - 420, he recorded that *Bunkulu*, the flowers of *Muyinga*, for which no taxonomic name is given, but





which he described as *a bushy, yellow-flowered papilionaceous herb*, were cooked with groundnuts in a porridge in famine years. The fruits of *Parinari* and *Uapaca nitida* were eaten regularly.

Trapnell's interlocutors also informed him about trading activities. For example, Muchila himself sold poultry (a shilling each) and cassava. Goats were sold at neighbouring Kalomo (European and African customers), and groundnuts were sold to nearby missions. Other non-agricultural economic activities were recorded such as the manufacture and sale of canoes and paddles (page 449), the processing and sale of tobacco (page 398) and the manufacture and sale of iron hoes (page 439).

Trapnell described the vegetation as he saw it on the traverse, but also recorded information from informants about the lateral extent of particular formations. For example, at Kafushi on the inaugural traverse, he was informed that the *I. paniculata* country extended 1.5 days north, followed by four hours grass on sandy soil, 4 days north-west, 1.5 to 2 days south. Trapnell estimated that one day's travel was equivalent to 15 miles. While not stated explicitly it would appear that this information was used to delineate vegetation map units in the map accompanying Trapnell and Clothier (1937), and in the final Vegetation-Soil map, perhaps with the cross-hatching rather than solid pattern to indicate the uncertainty.

## 4.6   Soil, vegetation and land use in the traverses records

Having considered Trapnell's methods thematically, we now focus on his observations in some of the fieldwork undertaken in the Upper Valley and adjoining Plateau in 1932. We then present summaries of observations in tabular and graphical form, and then examine what the traverse records show about relationships between soil and farming practices, and changes in the latter during the time of the Ecological Survey.

The distinct units of land which the Soil and Vegetation Map (SVM, Trapnell et al., 1947) delineated as Plateau Soils (P7: Southern *Isoberlinia globiflora – Brachystegia* woodland , P5: Central *Isoberlinia paniculata – Brachystegia* woodland) and Upper Valley Soils (U2: *Combretum – Afrormosia* and *Pterocarpus– Compbretum* transitional grass-woodland, U3: *Acacia – Combretum* thorn) were traversed early in the inaugural survey. For example, on 14[th] June 1932 the route from Lusaka to Broken Hill (now Kabwe), started over yellowish soils under *Brachystegia flagristipulata* and *B. hockii* (P7). African cultivation was observed in a valley with some fig trees. Later on that route 'buff' topsoil over deeper orange subsoil was observed under transitional woodland (*Afrormosia, Combretum, Albizzia and Terminalia* (U2) before passing into *Acacia campylacantha* (U3) thorn. Again, cultivation was observed in the transitional woodland, but no details were recorded by Trapnell.

The route the next day from Kabwe to Kafushi passed onto *Isoberlinia paniculata* plateau soils, with some *Brachystegia* species, and *Uapaca*. The soil varies from pure white sand to 'buffish' clay, and the laterite blocks and underlying laterite, characterisitic of the plateau, were observed. After 10 miles or so on the Plateau the route passed on to dense *Combretum* with tall grass, then *A. campylacantha* before passing into more open *Combretum-Terminalia* country, then *Acacia woodii* with *Hyparrhenia* grass and *A. woodii*-grass cover before an *A. campylacantha* belt.





Both these routes cut across the Plateau and Upper Valley environments, and showed the distribution of char-
acteristic transitional vegetation before the thorn soils of the latter. This is not commented on in the notes at this
stage, nor is the term 'transitional' used. Although Trapnell observed some cultivation on the thorn soils, no detail
is recorded.

The third day out of Lusaka (16th June 1932) was spent in a cycle reconnaissance around Kafushi on Plateau soils
under *remarkably pure and uniform* cover of *I. paniculata*. Some variation was seen with *B. flagristipulata* near
dambos, and *Uapaca kirkiana* on shallow soils over laterite. Trapnell also observed the vegetation characteristic
of the Upper Valley thorn soils (*A. campylacantha, Hyparrhenia rufa*) on 'sweet' dambos with good grazing, and
the transitional vegetation (*Combretum, Terminalia*) over the poorer 'sour' dambo.

Trapnell's observations on the around Kafushi included his first reference to farming practices on the Plateau,
but these are rather sparse and are not suggestive of detailed discussion with informants. He noted that trees were
lopped and burned, and inferred that shifts of the cultivated site were *probably fairly frequent*. He also noted that
the dambo slopes were cultivated, but not to the waterside, in contrast with dambos under *B. longifolia*. This
tree cover had been observed on previous days, although without observations on dambo cultivation in Trapnell's
notes.

The continued Ila-Tonga traverses in August/September 1932 after Adamson's departure and Trapnell's period
of illness were south of the Kafue (Figure 3). These covered Plateau and Upper Valley environments, as well as
routes across the sedimentary soils of the Kafue Flats. As noted above, Trapnell's descriptions of agricultural sys-
tems, alongside the ecological descriptions and comments on soil, become more detailed. For example, at Shin-
sana's village, visited on 17th September 1932, he described a Plateau setting under *B. flagristipulata* and some B.
hockii over gravelly or old cultivated soils. There were species of *Hyparrhenia* grass, including *H. rufa* on dambos.
He noted a fine sandy loam soil, chestnut to brown in colour and relatively shallow with an underlying layer of
ironstone nodules and quartzitic gravel which is charateristic of the old plateau soils, the layer being thicker near
the dambo. Within this ecological setting he noted that the community were engaged in bush cultivation. Sites
were selected on the presence of *H. filipendula* and *B. flagristipulata*. Opened land was cultivated for five years,
with new land opened each year for the cultivation of groundnuts. They returned after a four-year fallow, and the
site was then abandoned. In addition he noted that the community was vulnerable to famine in dry years.

The next day, 18th September 1932, Trapnell made a similarly detailed set of observations at a site, Muchila's
visit, on the Upper Valley. The dominant vegetation was what Trapnell referred to as Transitional, *Afrormosia –
Combretum* over soil derived from granite, and some of the sites under cultivation were under *Acacia campylacan-
tha* characteristic of the thorn soils of the Upper valley beyond the transitional fringe. This community cultivated
*Afrormosia* bush, selecting sites with *Afrormosia* and *H. filipendula* for groundnut crops. Sites under *H. filipen-
dula, A. campylacantha* and *Combretum* were selected for growing maize and sorghum. Land was cultivated for
four years, with millet grown in the third year, after a four year fallow the community returned to the site which
was then abandoned.



A similar level of detail was provided at many sites in the remaining traverses in the vicinity of the Upper Valley. The observations are summarized in two figures. The vegetation species named at cultivated sites (excluding the observations in Reserve IX) are shown in Fig. 6. Because of the purposive nature of sampling on the transect the relative proportions of these species should not be treated as evidence of the vegetation species associated with cultivation in the Kafue Basin at the time of the surveys, but they do show the picture provided by the Traverse Records as subsequently interpreted by the survey team. Fig. 7 shows the number of references to particular species, or associations of species, as indicators of the suitability of a site for selection for cultivation, as recorded by Trapnell from discussions with informants.

Trapnell's observations of agricultural systems in these traverses are compiled in Table S4 in the supplementary material. These and similar observations, along with those made by Clothier across the Kafue Basin (Clothier, 1933), are generalized in the descriptions of agricultural systems provided by Trapnell and Clothier (1937) and subsequent reports. We have noted above some of the challenges in the interpretation of the accounts of farm systems in the traverse records. Nontheless, they contain a wealth of detail on practices following site selection with respect to land preparation (for example, the burning of tree branches and other biomass), cropping sequences, the extension of cultivated land in successive seasons (e.g. by planting groundnuts in extensions on the plateau), variations in the size of cultivated areas (reflecting soil quality) and the spatial complexity of cultivation (with certain crops grown in ash heaps, some on garden margins and some on dambo margins).

One notable feature of Trapnell's observations on agricultural systems is the information he inferred or gathered directly from informants about recent change in farming practice. Again, these are compiled from the Upper Valley traverse records in Table S5. At three locations, visited in February 1934 in the vicinity of Monze south of the Kafue river, Trapnell was told how maize as a crop had supplanted sorghum. Sorghum itself appeared to have replaced millets as the dominant crop. At one site bullrush millet had preceded sorghum, and others bullrush millet and finger millet. At one site, Benzu's (23rd February 1934) the demise of sorghum was linked explicitly to the arrival of Europeans. At two sites on adjoining Plateau maize was the dominant crop at the time of survey, in some cases in combination with sorghum. At one site this was linked explicitly to the railway, where the maize was taken for sale. At a third site maize, sorghum and finger millet were planted, the latter on anthills. It was stated that previously the crops had been bullrush millet and sorghum. Trapnell noticed that the practice of growing bullrush millet and sorghum had changed where ploughs were used, with alternate rows of the crops grown rather than separate gardens. He also noted that bullrush millet had spread as a crop onto sandier soils in the Zambezi catchment.

The introduction of the plough was a critical technological change which was taking place at the time of the Ecological Survey. As noted in the previous paragraph, Trapnell observed change in farming of millet and sorghum through the use of ploughing, but it was also associated with increased cultivation areas and extended periods of cultivation of land for maize production in response to markets opened by the railway (see observations in the Pemba – Kalomo road traverse of 1933, page 528 et seq.) The traverse record, however, provide rather limited detail about the use of the plough, by comparison, for example, with traditional technologies such as soil selection





or burning for ash fertilization. There is one incidental reference in the Ila-Tonga series (page 525, where it was
noted that certain soils under *Acacia* could be ploughed before the rain. and one reference in the Sala Reserve field
notes (page 449) at a site where black clay soil in alluvium was ploughed only in higher (better drained) locations.
There were more (four) references in the Road Traverses, where ploughing downslope was observed (page 529),
and where Trapnell noted that the introduction of the plough at a Plateau site had resulted in intercropping of
sorghum and bullrush millet on alternate rows where previously they had been grown in separate adjacent gardens
(page 529). This seems to be the only observation involving ploughing in these particular traverse records which
notes specific technical information as to how the plough was used in the system. In the Upper Valley at Mapanza
mission he noted that Balundwe people had moved from cultivated Transitional bush sites to the riverbank, using
ploughing, which he associated with *breakdown of soil* noting that erosion was general.

### 4.7   Trapnell's field observations and the 'ecological concept of development'.

Speek (2014) treats Trapnell as a key figure in the emergence of an 'ecological' theory of development in Zambia.
Under this account, the developmental trajectory of a local ecosystem, incorporating an African 'tribal' group of
cultivators, was either adapting towards some stable 'climax' state, analogous with a primary vegetation succes-
sion, or degrading. The African cultivator was not granted conscious agency in this model. Ideally the cultivator is
operating in harmony with nature, in contrast with the 'defeat of nature' by European cultivators. This theory was
seen as a reason for separating African and European cultivators, for example through Maize Control regulations
to avoid direct competition for grain markets.

Moore and Vaughan (1994) note that the variations to be found within agricultural systems in the Zambian set-
ting were often interpreted in ethnic and evolutionary terms in which a particular group had developed a system
along some trajectory, often with an additional narrative of a contemporary breakdown of the system, for example
as a result of large-scale labour emigration. However, they comment that Trapnell (1943) was sceptical about such
interpretation. Does our reading of Trapnell's field work support the interpretation of Speek (2014)?

The traverse records in the Upper Valley show Trapnell identifying communities as 'backward' if they were found
to be cultivating poor soils, or undertaking soil selection in ways which fell short of a paradigmatic ecological prin-
ciple. At Benzu's village (page 552) Trapnell noted that the community selected land with long grass, but concluded
that they were *unconscious of their practices*. Similarly, he noted that Munampelo's people (page 467), while select-
ing sites with tall grass, where Sorghum would grow well, did not know names for the grasses, and compared them
with *true Ba Ila* who know the grass names. Here the degree of 'consciousness' of an African cultivator seems to be
measured by how far they approximate to a scientific ecologist, although it is not clear that Munampelo's struc-
tural classification of vegetation is any less effective for the cultivator than a taxonomic one. In this respect, and
in his idea that even the successful African cultivator was selecting soils from plant species or vegetation types
*intuitively and without conscious thought* (Trapnell, 1937), Speek's identification of Trapnell with the ecological



theory of development has some force. However, Trapnell's observations, particularly his field records, do show a more nuanced understanding, as we now show.

Trapnell knew that the state of affairs in the Upper Valley and surrounding landscape was more complex than
630 a picture of different ethnic groups, adapted to differing degrees to their local environments, and part of local ecosystems either stabilizing or degrading. First, he understood that particular villages or wider communities, following a common set of soil selection practices might comprise more than one ethnicity. At one site, with *Acacia albida* as the dominant tree, he noted *People here mixed: Batonga and Balundwe* (page 385). He was also aware that communities moved on the Plateau and Upper Valley in response to varied factors. He encountered one Tonga
community at a site they had occupied for 7 years, after leaving a reserve (page 560). He also identified cultivation practices which were not accommodated by a simple classification, noting types *intermediate between the Tonga "circle" cultivation of the plateau bush and the differentiated bush/dambo head or associated dambo cultivation of Transitional bush,* (page 566). Furthermore, Trapnell and Clothier (1937) observed in the final report for Central and Western Zambia that movement of 'tribal' groups led to change in agricultural practices independent of
any European intervention (paragraph 188). These changes in practices might be through direct adoption of the methods of new neighbours, or through adaptation of neighbouring systems to create a new one (a groundnut-maize-millet rotation developed by Ila cultivators on sandy soils).

Trapnell was also aware of the contingencies which cause communities and people to move, and recognized cases where cultivation practice changed when a community moved to a new environment — in the Central and
645 Western Report, Trapnell and Clothier (1937), he noted that Lamba-Kaonde people on the Northern Plateau had changed their agricultural practices where they had penetrated to the Southern plateau (paragraph 90). Nonetheless, he described them as *backward .., lacking in crafts and primitive in diet, sowing [Sorghum] broadcast in ash-fertilised land.* In contrast (paragraph 91) the Ila and Tonga *tend generally to a higher level of agricultural development,* by which he appears to refer to features of the Southern plateau system, and cultivation of maize on the
650 Thorn soils of the Upper Valley, not necessarily criticizing ash cultivation as such which was known to counteract soil acidity as well as providing plant nutrients.

It has been noted the the Lamba people, with their origins in the Copperbelt region, were widely held in low-regard from the 1930s Colony to post-independence Zambia (Siegel, 1989). This has been ascribed to their resistance to Colonial models of development, not least in preferring agricultural activity to paid work in the mines.
Siegel (1989) shows that, for some, this was a deliberate ideological choice, the 'African Watchtower Movement', based among the Lamba, actively rejected urban life as a colonial innovation. Others were simply happy to engage with this new order from the margins by selling agricultural produce, or engaging in mine work for short periods only. From the colonial perspective, including Trapnell's, the Lamba appeared unambitious, and therefore backward. Siegel (1989) notes the difficulty of reconstructing the Lamba perspective, but their disengagement, or min-
imal engagement, with the colonial economy appears to have had roots in a strong sense of grievance at the loss





of land to mining activity, a suspicion that the colonial authority's schemes aimed to dispossess them further, and the perception that the colony's administrators had interfered unjustifiably in the role of their traditional leaders.

The Ila-Tonga in general receive a positive assessment from Trapnell, and their use of the Thorn soils of the Upper Valley enabled rapid agricultural development, which he observed, along with emerging problems, in the Sala
Reserve and the vicinity of Pemba. Mobility in space facilitated this development, particularly in taking up land with access to the railway line and the maize markets which it served. Colson (1962) was to note, in field work in the mid 1940s, that Tonga communities showed particular mobility on the southern plateau with a minority of people living in the village of their birth, and a general tendency to move westward over the lifetime of the individuals she recorded. This mobility was facilitated, in part, by distinctive historical and social factors in Tonga life, which
meant that the Tonga could transfer allegiance between traditional headmen with relative ease. Cultic considerations were also important. For the Ila, ancestral spirits were associated with sacred funeral groves on particular areas of land which therefore held an ongoing meaning and a tie to the location, but for the Tonga obligations to ancestral spirits were met at domestic shrines, either at the doorway or central pole of the hut (Jaspan, 2017).

In summary, Trapnell's field records and their synthesis in the final reports, suggest a more nuanced understand-
675 ing of the development, sharing and adoption of cultivation strategies than is perhaps consistent with Speek's (2014) ecological model of development. Still less is Trapnell's understanding consistent with the account of ecological survey given by Anker (2002) in his account of Bourne's contribution on air photography as a survey tool to the Fifth International Botanical Congress in Cambridge. Anker (2002) states (page 134) '*The political aim of ecological sampling in a grand survey of the empire was thus to find environmental solutions to social unrest among*
680 *diverse human ecological groups in the colonies. The idea was to divide different races according to their corresponding ecological zones.*' This is hardly reflected in Trapnell's understanding of how peoples moved between environments, and adapted to them. Nonetheless, Trapnell's perspective on African cultivators, while sympathetic, also shared the limited colonial understanding of the factors other than agronomic and economic, including cultic, ideological and political concerns, which might motivate their decisions on what crops to grow, and where to
685 grow them.

## 5 Early syntheses

In this section we consider the early outputs from the Ecological Survey, and the syntheses which they present on the soils of the Upper Valley and surrounding plateau, the traditional practices of cultivation, and the challenges for development.

### 5.1 Clothier's report on the Kafue Basin

The first output from the Ecological Survey was a report by Clothier on observations from 1932 – 1933 (Clothier, 1933). Unlike Trapnell, Clothier was an agriculturalist, and a recent graduate from the Imperial College of Agri-





culture in Trinidad (Tilley, 2011) but the report sets the agricultural observations in an ecological context which is entirely consistent with both Trapnell's field notes and the subsequent published reports detailed in section 6. Although consistent, Clothier's ecological terminology is somewhat different from that of the final reports. He identified three 'bush types', the Plateau bush predominantly on the old peneplain with *Brachystegia – Isoberlinia* vegetation and fringing *Combretum – Terminalia* tree/grassland; Transitional bush on residual sandy soils, i.e. at the margins of the plateau where rejuvenation increases the relief and soil fertility, and with *Combretum – Aformosia/Albizzia* scrubland; Sweet Bush on colluvium and lower plains of drainage basins with *Acacia* tree-grassland and tall *Hyparrhenia* grasses. 'Sweet bush' denotes pasture land over soils with a large nutrient supply relative to the rate of primary production – 'soetveld' in Afrikaans in contrast to 'sourveld' (Ellery et al., 1995).

Clothier (1933) uses this framework to describe a set of 'cultivation systems'. These are set out in some detail in Table S6 in the Supplementary material. Certain systems are characteristic of particular bush types. For example, Residual Cap cultivation was found in Plateau bush where communities had limited or no opportunities to cultivate in dambos. This system was limited by the nutrient supply from cut and burned vegetation, and continuous cultivation was limited to two or three years with millets and gourds the principal crops. Similar systems, but typically with longer cultivation periods, were found in Transitional bush (Dense scrub system) mainly cultivated for maize. In the Sweet bush, sites were cultivated for substantially longer (up to 10 years in *Acacia woodii* belts and *A. woodii – A. campylacantha* transitions. Dambo heads, and Sweet Dambo sites in Plateau Bush, with *A. campylacantha* and *H. rufa* or *H. filipendula* were fertile and productive sites, contrasting with the dominant local soil and vegetation and Sweet Dambo cultivation was also found in the Transitional Bush.

Clothier also noted that choices on cultivation often depended on a range of soil and bush types available to a community. Residual cap cultivation, for example, would be found in Plateau bush only where a community did not have the option of dambo cultivation. Similarly Colluvial Belt cultivation would be practiced only for subsidiary gardens in Transitional bush where communities had the option of cultivating in Sweet dambos. Furthermore, communities near the railway line had the opportunity to sell maize into larger markets for cash, and this influenced decisions on land use. For example, on Thorn Fringe sites with *A. woodii* in Transitional bush, large maize gardens could be found where communities had market access, otherwise maize was grown on smaller plots in such sites, along with groundnuts, cowpeas and gourds. In short, Clothier's overview emphasizes that a community's decisions about cultivation took account of ecological conditions over a range of accessible sites, as well as opportunities beyond subsistence production.

## 5.2 Trapnell's contribution to the Second Meeting of African Soil Chemists, Zanzibar

On the initiative of William Nowell, Director of the Amani Research Station, Tanganyika, the soil chemists from British East African Territories convened in Amani in 1932 to discuss, primarily, the production of a soil map of the region (Milne, 1932). A second meeting to discuss progress was held in Zanzibar in 1934 (Milne, 1935). Trapnell had just begun his fieldwork in Zambia when the East African soil chemists met at Amani. By the time of the





meeting in Zanzibar he had begun a correspondence with Milne, had submitted three abstracts to the meeting and been engaged to open a discussion entitled 'Ecological survey in its relation to soil survey'. However, Trapnell did not attend, a letter from Milne to Trapnell after the meeting (Milne, 1934) shows that this was a decision of the Department of Agriculture in Northern Rhodesia, on financial grounds. Nonetheless, Trapnell's contributions appear in print in the proceedings (Milne, 1935), and an abstract entitled 'A vegetational grouping of soils in Northern Rhodesia south of latitude 15° 30′' is his first printed account of the principal soil groups of central and western Zambia, and their relation to vegetation outside the pages of reports of the Northern Rhodesian government. Additional notes received from Trapnell during the meeting, accompanying draft map sheets, were read out.

As Trapnell did not present this in person, there is no amplification of a fairly terse abstract, neither is there any reported discussion. The abstract's stated aim was to describe *the main visible characters of soils grouped according to the principal vegetation formations, using, as far as possible, the groupings of Henkel's vegetation map of Southern Rhodesia [Henkel (1931)] together with what has been recorded of the vegetation of Nyasaland and Tanganyika.* Trapnell set out four 'main groups' of soils, with subgroups, stating that *main groups answer approximately to soil groups of different history, the sub-divisions answer to fertility distinctions.* Trapnell indicated that this classification should allow comparisons with East Africa, and postulated that similar soil–vegetation relations might be found elsewhere at comparable altitudes and where the rainfall is similarly unimodal.

Trapnell's main groups, as presented to the Zanzibar meeting, are shown in the first column of Table (2), with the corresponding units from the subsequent survey reports and the Soil–Vegetation map (Trapnell et al., 1947). Here we focus on the Plateau and Upper Valley groups and their subgroups.

The *Plateau group* of soils was characterized by *Brachystegia–Isoberlinia–Uapaca* tree cover on the archaean complex. The soils were described as eluvial cover of the ancient peneplain where this has not been covered by Kalahari sand or rejuvenated by recently renewed erosion cycles. Characteristic of the soils are nodular ironstone deposits at level sites, either in the profile or exposed at dambo margins or by rejuvenation.

Trapnell listed four subdivisions of the Plateau group, and the abstract gives no details beyond the names. They pick out contrasts in soil texture: *Sandy plateau soil* and *Plateau red loams*, although the latter were identified as a new main group in the North-Eastern survey (Trapnell, 1943) and the final map (Trapnell et al., 1947). The *Shallow nodular soils* and *Ironstone swamp soils* pick out local conditions related to drainage and erosion history with clear implications for land use.

The *Upper valley group* of soils has distinctive vegetation: *Combretum* and *Acacia* tree cover on the archaean complex and on younger sedimentary rocks other than those of the Karroo group found in the lower valley. The soils were formed as a result of erosion on the ancient peneplain, resulting from rejuvenated drainage, and so appear less mature than the Plateau soils. Ironstone concretions are absent, or present as residual decomposing surface blocks.





More information is provided on the subdivisions of this main group than for the Plateau soils. The *Transitional soils* are residual or residual-colluvial soils (i.e. soils formed either *in situ* at denuded sites, or in a mixture of such residual material and colluvium. They comprise *Immature sandy loams, Grey colluvial soils* and *Red sandy loams.* These are contrasted with the *Thorn soils* with three subdivisions:*Red thorn loams, Black thorn clays* and the *Winterthorn alluvium* (Winterthorn = *Acacia albida*). At this stage Trapnell did not name characteristic vegetation of the Transitional soils, and it is left implicit that the thorn soils were predominantly formed in colluvial or alluvial material. The abstract promised further notes on the fertility of these soils and their suitability for staple African crops, and, had Trapnell presented these, they would presumably have been consistent with Clothier's report (Clothier, 1933).

## 6 Reports of the Ecological Survey

### 6.1 The Soils, Vegetation and Agricultural Systems of North-Western Rhodesia, Trapnell and Clothier (1937)

This was the first publication from the Ecological Survey, apart from Departmental Reports. Some of the key information contained in this report about soils and the vegetation mapping units is summarized in Table S7 in the supplementary material.

### 6.1.1 Soils and Vegetation

The account of the soils in the report is structured mainly by topography, reflecting Trapnell's summary for the second East African meeting (Milne, 1935). This is presented as an alternative to a description of the soils based on chemical analyses, which were not available *due to retrenchment of the Soil Chemist position.* In the account of the soil classes the recently published East African Soil Map (Milne, 1936) is treated as normative, the point is emphasized that the soils are described *in terms consistent with those employed by the recently-prepared East African Soil Map.*

Here we focus on the Upper Valley soils and their neighbouring Plateau soils. Trapnell and Clothier (1937) note that the latter are widespread on the south-central African plateau, and, while showing some variations with respect to colour and particle size distribution, have in common that they have formed on topography in a state of maturity, and those stability, in which they have been subject to seasonal leaching over a long period of time. Typically nodular or concretionary ironstone is found close to the regolith, and this is most pronounced in poorly-drained conditions which may arise from flat topography, impervious underlying rock or proximity to a dambo.

A broad distinction was made between the Northern Plateau (north of the 40" isohyet), deeper soils with a larger clay content and brighter colours than these of the Southern Plateau which typically are 50 – 60 % sand. Chemical analyses were available only for the southern soils, and these indicated low fertility.





Three soil subgroups were recognized. The first are Older Ironstone Soils, pallid and shallow, from the older land surfaces with little variation despite the underlying geological variation, and with ironstone which drew a parallel with the Murram soils of Milne (1936). While used to grow finger millet, these soils were described as *agriculturally useless.*

Light-Coloured Plateau soils were associated with partially regraded plateau surfaces, and show greater variation both with climate expressed in colour variation from yellow and orange clay soils on the Northern Plateau, orange and pink to buff soils around the Copperbelt and sandy soils of more muted colour on the Southern Plateau. These soils are explicitly compared to the Plateau soils of Milne (1936). Finally, Red and Brown Plateau Soils were identified, your soils in residual or colluvial material, including deep-red clay soils over calcareous parent material on the Northern Plateau. Theses soils are correlated with the Red Earths of Milne (1936),and are described as including the most fertile Plateau Soils.

    Trapnell and Clothier (1937) mapped four principal units on Plateau soils. On the Northern Plateau they identified *Brachystegia* woodland on clay soils, and *Brachystegia – Isoberlinia* woodland on more variable soil. On the Southern Plateau *Isoberlinia paniculata – Brachystegia* woodland was mapped over sandy soils and *Isoberlinia*

*globiflora – Brachystegia* over sandy loam, extending of the Plateau onto adjacent Kalahari sands, the 'Kalahari contact' soils. The map legend also groups *Isoberlinia globiflora – Brachystegia* woodland over the escarpment hills, extending to the lower valley with the Southern Plateau units.

    The Upper Valley soils are contrasted with their neighbouring Plateau soils. As noted in section **??** Trapnell compares them to the non-calcareous Plains soils of Milne (1936). It is noted that the distinction between the Upper

Valley and Plateau soils was first recognized because of the former's distinctive vegetation cover. However, the fundamental difference lies in the Upper Valley's more modified topography, the country being broken or rolling, rather than graded to a mature surface, and with free drainage. There are also some differences in parent material from the Plateau Soils, with limestone and mica schists common. The younger soils, both residual ones formed in situ on rejuvenated surfaces and those formed in resulting colluvium, are loams in texture, varying from sandy to

clay loams. While they might show some mobilization of iron as mottles, or coatings on rock fragments, ironstone formations are lacking. Like the soils of the Lower Valley, the subsoils may have a basic reaction. The key practical difference from the Plateau soils is their larger base saturation and content of phosphate and nitrogen, making them notably more fertile.

    The key subdivision of the Upper Valley Soils, made by the time of the Central and Western report, was between

820 the Thorn soils under *Acacia*-dominated cover and the Transitional soils, intermediate between the Thorn soils and the Plateau types, although the vegetation map published with Trapnell and Clothier (1937), did not attempt these as mapping units.

    The Transitional soils had tree cover dominated by *Combretum* and members of the *Papilionoideae*, notably *Afrormosia angolensis*. The Thorn soils were mainly on colluvial sandy loam material, generally more coherent

than the Transitional soils, but also included some alluvial soil under *Acacia*. The Thorn soils were described by



Trapnell and Clothier (1937) as the *best maize land and dry grazing in the country*, generally with a larger nitrogen content than other soils, the phosphate content being variable. The Transitional soils, mainly residual, were described as well-drained, friable sandy loams of variable coherence and with double the phosphate content of the adjoining Plateau soils. They were regarded as light maize soils with potential to grow tobacco and cotton.

This synthesis inevitably requires generalization of the observations made on the ground, and recorded in the Traverse Records. For example, in the Macha to Namwala record, commencing on page 533, there is an interval, mapped to Transitional soils in the Upper Valley, where the records show a complex pattern of woodland, tall grassland on level ground and gentle slopes with red soils over schists and quartz. Characteristic species of the Transitional soils were seen (*Albizzia, Pterocarpus, Afrormosia*) along with other tree species (*Afzelia, Ostryoderris*

and even some 'rogue' *Brachystegia*. Of particular note is the appearance of *Acacia campylacantha* and *A. albida* on dambo soils surrounded by *Brachystegia of the Plateau* (e.g. at Mukulaikwa's, page 438). Alluvial dambo soils carry local vegetation characteristic of the colluvial Thorn soils in the Upper Valley, and also have considerable agricultural value. This was made explicit by Clothier (1933) in his Kafue basin report (section 5.1) where he notes the parallels between the Type C cultivation systems of the 'Sweet bush' under *Acacia campylacantha* and the

Dambo Heads and Sweet Dambo of the Plateau Bush where gardens were established where *A. campylacantha* was found along with tall *Hyparrhenia* grass.

### 6.1.2    Agricultural systems

Trapnell and Clothier (1937) describe five principal agricultural systems of the Upper Valley, and two principal Southern Plateau systems, along with local variants. The descriptions of these systems are summarized in Table S8

in the supplementary material. The three Transitional Country systems are distinguished with respect to topography (dense scrub adjoining the Plateau, Dambos and their margins and Bush gardens. In all of these ploughing might be practiced, and the cultivation period extended beyond that under traditional cultivation. A single Thorn soil system was recognized, often subject to large-scale cultivation with the plough, although not by the Ila people. A transitional sand system was also described, dependent on burning of brushwood, and cultivated to maize in

the first year, then to bulrush millet with groundnuts planted in garden extensions in the second year. Sometimes followed by maize with or without sorghum.

The primary distinction made within the Southern Plateau system was between *Isoberlinia paniculata – Brachystegia* woodland, cultivated as Main gardens and Village gardens in the central regions and over poorer Kalahari contact soils, and the *Isoberlinia globiflora* woodland cultivated by Tonga people in the south, again in main gar-

dens and village gardens. All variants were dependent on the felling, piling and burning of tree branches followed by hoeing of all the cleared land.

As with the soil and vegetation observations, the agricultural systems delineated by Trapnell and Clothier (1937) are generalizations of the complexity which they observed in the field. For example, at Chongo's (31st August 1932, page 381) Trapnell observed what he called 'semi-permanent' cultivation on Thorn soils of the Upper Valley, with



three years cultivation, followed by two years' fallow practiced on two fields, with the second one cultivated for the first time in the third year of cultivation in the first, compensating for reduced yield. Trapnell called this a system of 'minor shifts' (i.e. of the principal field in cultivation), with major shifts happening perhaps only when a son took over cultivation from his father. The three-years cultivation might be extended to four where less land was available.

Similarly on the plateau, at Chisako's (1934, page 542) under *Isoberlinia paniculata* Trapnell observed a complex variant of the Southern Plateau system where grass was brought in to supplement the wood which was burned around an anthill, and where earlier-maturing crops were grown on the edge of the plot, and where the larger sites we sown with different sequences of maize, finger millet or pumpkin depending on local soil conditions, which might be followed with a sweet potato crop before being abandoned.

### 870 6.1.3 Agricultural Development

The Ecological Survey Reports, (Trapnell and Clothier, 1937; Trapnell, 1943) provide recommendations for agricultural development, structured around the traditional systems which had been identified. Trapnell and Clothier (1937) comment that improvement of *a consistent but flexible body of agricultural tradition ... is not a task to be undertaken lightly*. They also note that the Ecological Survey can be regarded only as a first attempt to develop the 875 understanding of these systems which is needed prior to any attempt at improvement. In the case of the Upper Valley System, however, they observe that, in the vicinity of the rail line the main priority is remediation in the light of rapid change which has already occurred. However, they see the traditional Upper Valley system as offering the best basis for development. Tentatively they suggest that some changes to rotations, including groundnuts, cotton as a new crop and the use of composts could be preferable to increased cultivation of dambos, or of thorn soils 880 with greater potential for European agriculture.

### 6.2 The Soils, Vegetation and Agricultural Systems of North-Eastern Rhodesia, Trapnell (1943)

The traverse records on which we focused in this study contributed to the report of Trapnell and Clothier (1937), and so here we focus in brief on the emerging structure for representing soil variation as it stood after completion of field work across the country.

The account of the Regional Soil Types starts with more reflection on general principles than did the earlier report. Three primary factors are identified, to which soils owe their characteristics. These are climate (past and present), parent material and the age of the land surface and nature of the changes that have taken place in relief. The latter is key in determining maturity of the soil and the extent to which past or present climate influences soil properties. These effects, says Trapnell (1943) *cut across the broad zonal arrangement of climatic soil types*. This 890 emphasis on a climatic pattern is in contrast with the North-Western report where soil variations attributable to climate (e.g. the strongly alkaline soils of the Lower Valley) do not map simply onto the emphasized topographic grouping. This is most probably because extension of the survey to the east of the country introduced a substantial





region of lower latitudes than those traversed in the west. Trapnell (1943) notes, for example, the pronounced contrast between the humid environment in which the grey-humic soils of the Lake Basin region were formed, and the 'pedocal' conditions in the Lower Valley environments where intense evaporation and soil moisture deficit results in the development of alkaline soils, sometimes with nodular lime. Traverses in the east of the country also covered red earth soils correlated with those of the East African Soil Map.

Of particular interest here, however, is the comment of Trapnell (1943) on how age of the land surface, and changes of relief, modify the effects of climate and parental material, because this is key to the genesis of the Upper Valley and its distinctive and agriculturally important soils. The rejuvenation of the ancient land surface creates residual and colluvial soil parent material, within which soils develop under near-contemporary climatic conditions. Trapnell (1943) describes the Upper Valley soils as *essentially soils of the present, formed, and in North-Eastern Rhodesia probably still in process of formation, in areas rather lower than the general plateau level where the land surface appears to have undergone comparatively recent modification.* This younger parent material is one reason why the Upper Valley soils are typically very fertile.

As in the previous report, Trapnell explains the use of a *regional physiographic basis* for classification on the paucity of soil analytical data. However, he states that, when physiographic differences are accounted for he expects to see a 'climatic sequence'. Such a sequence might be seen in the transition, north to south, from Plateau soils, *Light pink-brown or buff topsoils over "rawer" coloured subsoil. Largely structureless save for ironstone pellets* to Upper Valley Soils: *Warmer chocolate-toned' soils with increasing clod-structure to brown pedocal soils with vertical cracking in the Lower Valley.*

### 6.3 The Vegetation-soil map of Northern Rhodesia

The notion of climatic sequences cutting across physiographic differences, introduced in the North-East report (Trapnell, 1943) was developed further and presented in the memoir to the 1947 vegetation soil map (Trapnell et al., 1947). In the introductory paragraph (15) the 'climatic sequence' model of national-scale soil variation is developed and extended. In the North-East report the Lower Valley Soils, Upper valley soils and Plateau soils were treated as a sequence from those formed in the most wettest conditions (plateau) to those formed in the most arid conditions (pedocals of the Lower Valleys). This sequence was recognised *after physiographic differences were accounted for.* In the 1947 memoir the Red Earths and Lake Basin Soils (or Grey Earths), were placed at the top of a "main series" (after the Plateau soils), representing, respectively tropical and more temperate high-rainfall conditions, the former being lateritic (in the sense of including pisolithic or concretionary ferruginous material) and the latter humic and podsolic. Two associated series are identified, again on a dry-to-wet climatic gradient. The first are "hydrogenic soils", from black calcareous clays at the dry end through grey dambo soils to moorland and swamp peats at the wet end. The second are the "lithological types", essentially soils on sand, from transition soils to Kalahari Sands to the Bracken Sands in the wettest conditions.



Within this fully developed framework Trapnell et al. (1947) notes that the Plateau soils give way to Upper Valley soils in lower areas of younger relief. The soils of the Upper Valley are described as *warmer-toned* pink-brown or cocoa-coloured to chocolate/darker brown soils with a more pronounced increase in base saturation and exchangeable bases with depth than on the neighbouring Plateau soils. The earlier correlation with Non-calcareous Plains Soils of Milne (1936) is reiterated.

Trapnell et al. (1947) notes that there are associated "limited belts" of soils with affinities for red loams, treated as intrazonal soils. In 1962 the 1947 map was reprinted, and it is from this that the map published by Smith and Trapnell (2001) was produced. One difference between the two maps is the introduction of some red 'R' symbols denoting the occurrence of red loams on areas mapped to Upper Valley soils in the eastern map sheet, and easternmost parts of the western map sheet. These would have been introduced by Trapnell, who by then had worked in East Africa.

## 7   Conclusions

Trapnell and Clothier (1937) offer a framework for thinking about soil variation in the Upper Valley and surrounding plateau of western and central Zambia which is based primarily on physiography. Trapnell presents a sophisticated understanding of how normal erosion, in response to a change in base level, and the consequent rejuvenation of the plateau, produces both residual and colluvial parent material for pedogenesis with a larger content of weatherable minerals and so greater fertility than the soils of the plateau. Furthermore, the climatic steady state to which this soil is converging is different from that reached by the Plateau soils in a contrasting palaeoclimate. These pedogenetic differences account for the observed ecological variation, and the associated differences in land capability for farming. In the wider west and central region there are climate differences related to topography between the Lower and Upper Valleys and the Plateau, and to the west a covering of aeolian Kalahari Sand imposes a parent material over the underlying bedrock, the characteristics of which vary with its thickness.

At the time of Trapnell's field work soil scientists such as Marbut, influenced by the Russian school of pedology through the writings of scientists such as Glinka, which Marbut translated from German to English (Anonymous, 1930) were convinced that soil conditions were primarily determined by climate. Differences between soils in a common climatic setting, inherited from parent material, could simply be attributed to the soils' immaturity. This view is expressed by Shantz and Marbut (1923) and provided the basis for their proposed map of African soils based on a handful of samples interpreted with respect to a climatic map. This school of thought was largely rejected by soil scientists working in British territories in Africa (Milne, 1932), and it is clearly not consistent with Trapnell's recognition of the importance of the effective age of the weathered material in explaining the ecological and agricultural differences between most soils of the Plateau and those of the Upper Valley.

On the extension of the Ecological Survey to the rest of the country (Trapnell, 1943; Trapnell et al., 1947), Trapnell encountered a wider range of climatic variation and so his overall pedogenetic model was extended. Trapnell



thinks of parent material, hydrology and relief as 'cutting across' the climate trend, so that soils under comparable
climates do not necessarily converge when other factors of soil formation operate at different spatial scales. In
recognizing that plural and connected factors control pedogenesis, Trapnell's practice in the field anticipated the
contribution of Jenny (1941).

This is a striking parallel with the views of Trapnell's mentor, A.G. Tansley, in Tansley's opposition to Clements's
climatic 'mono-climax' model of vegetation ecology (Van Der Walk, 2014). John Phillips argued for the mono-
climax model in southern Africa, specifically in dispute with A.P.G. Michelmore who noted the distinct vegetation
found at the margins of the central African plateau where it is rejuvenated by drainage — he describes what Trap-
nell would call transitional bush, (Michelmore, 1934).

Trapnell's work in the Upper Valley, and the overall Ecological Survey, is arguably of greater significance than the
East African Soil Map. While the work of Milne (1936) was an impressive piece of synthesis, it was largely a desk
exercise in correlating existing observations with a set of classes acceptable across the region. Milne's important
field observations, (Milne, 1947), were made after the map was published. By contrast, Trapnell developed a con-
ceptual model in the field which allowed the interpretation of the observed landform and vegetation to guide the
delineation of mapping units.

The geomorphological understanding behind the Upper Valley soils was at least as sophisticated as the catena
model presented alongside the East African Soil Map. Furthermore, the idea of regular lateral patterns of soil con-
ditions had already been identified by Henkel (1931), and Trapnell's field records frequently capture such lateral
patterns of soil and associated ecological variation in cross-section diagrams. That these were confined to field
notes and did not feature in published reports, meant that Trapnell's innovative practice of science in the field
was not recognized.

Trapnell states (Trapnell and Clothier, 1937) that the Ecological Survey soil units are consistent with those of
Milne (1936). The reading of his field records makes clear that they were not simply derivative from the East African
work. Emphasizing their consistency would have been important for validation of the Ecological Survey work,
given Bourne's critical comments (section 4.4), but may have resulted in an underemphasis of the originality of
the work in Zambia.

While Trapnell made only limited use of air-photography, his collaboration with Robbins showed how careful
field interpretation could be combined with imagery to support ecological soil mapping. Key to Trapnell's ap-
proach was the development of a conceptual model linking landform, soil development, vegetation and agricul-
tural potential. Studies have shown the model to be robust (e.g. Mukumbuta et al., 2022a), at least in so far as the
mapping is consistent with later work. Substantial loss of natural vegetation means that Trapnell's original frame-
work is no longer directly applicable in the field, but his approach offers a model for assessing challenges for land
management within a framework based on process understanding.

The wider value of Trapnell's field method was that it enabled him to identify ecological soil selection rules
used by African cultivators and to relate them to both farming practice and underlying soil variations, themselves



with a basis in physiography and climate. This provided the empirical basis for an understanding of the ecology of traditional farming practice, and for a representation of how these practices might be distributed in space. But Trapnell's understanding of the practices of African cultivators and their distribution was not simple determinism. He recognized that cultivators often have to move, over varying distances and for reasons which may be political, economic or the result of environmental change. In that setting rules are adapted, and Trapnell and Clothier (1937) gave examples of how a cultivator in a new environment might adapt and combine features of systems used by their neighbours. Although Trapnell regarded the adoption of European technology, such as the plough, by African cultivators as detrimental, it was not irredeemable, (Trapnell, 1934a) and further adaptation was possible. In this, his understanding of African cultivation is more nuanced than the 'ecological model' proposed by Speek (2014) in which the African cultivator is essentially an unconscious 'natural' actor in the ecosystem.

That said, Trapnell's field records do show how critical aspects of life on the plateau and Upper Valley can be missed without an openness to local 'ways of knowing'. Trapnell regards cultivators' interpretations of ecosystems as 'unconscious' where they do not parallel the taxonomic practice of a western ecologist in 'naming' species, but it is entirely possible that a 'structural' rather than a taxonomic approach to ecological indicators could constitute important 'indigenous knowledge'. Similarly, Trapnell was inclined to disparage groups such as the Lamba (?) who were reluctant to participate in the changed agricultural economy under colonialism, or to move to agriculturally superior land, with respect to their 'backwardness' or lack of industry. As we have seen, this overlooks other factors which may influence decisions on land use, including social, cultic and ideological ones. This underlines the importance of an approach to indigenous knowledge of soil and landuse which starts with careful and respectful attention to the conceptual framework in which the knowledge is produced. This requires cross-disciplinary research, particularly with linguists as illustrated by an early sketch of traditional soil classification in post-independence Zimbabwe (Nyamapfene, 1983), rather than looking for mappings of 'indigenous' soil classifications on to 'scientific' ones (e.g. Oudwater, 2003).

A target of recent decolonizing cultural and historical criticism is the colonial fallacy of 'emptiness' to justify the appropriation of land portrayed as unused, underused or misused (Wahu-Mūchiri, 2023). Trapnell's model of the complexity of environmental history on Zambia's Upper Valley and plateau, including climate change, fire, secondary succession under fallow and social adaptation to new conditions certainly could not sustain the fallacy, but rather, undermines it. As such it deserves wider historical attention.

The natural advantages of the Upper Valley made Southern Province Zambia's 'breadbasket' after independence, with maize production at over one third of the national total in the early 1980s (Kasali, 2011). However, in subsequent years these advantages have been lost, and the contribution of the province to national maize production had declined to around 10 percent by 2008. Kasali (2011) attributes this to the large-scale adoption of ploughing and stumping and attendant deforestation, precisely the concerns which Trapnell had raised. Kasali (2011) suggests that conservation farming strategies, drawing on traditional cultivation practices, offers a way forward. Stump (2010) argues that, too often, attempts to intervene to address such problems in Africa start from



judgements about land use, based on historical arguments which are unsubstantiated. Traditional practices are,
according to Stump (2010), typically framed in oversimplified terms as 'Ancient and backward' or 'Long-lived and
sustainable'. In the setting of the Upper Valley we are better-placed, thanks to Trapnell, to make more nuanced
historical judgements, and to recognize traditional practices as 'sustainable because capable of adaptation, and
of underpinning further adaptation.' This has implications for how future challenges can be addressed.

*Author contributions.* Funding acquisition was by RML, LC and SS. Systematic reading of Trapnell's field records was under-
1035 taken by IM, RML, NLM and MJH. Archival materials were located, read and reported by NLN, MJH and RML. Visualization of
data was by RML. RML, NLN and MJH produced the initial draft of the text and all authors contributed to subsequent revisions
and production of the final version.

*Code availability.* The data presented in this paper are available upon reasonable request to the corresponding author.

*Competing interests.* The contact author has declared that none of the authors has any competing interests.

*Acknowledgements.* This research was funded by the United Kingdom's Arts and Humanities Research Council [Grant No.
AH/T00410X/1] through their Programme Cultures, Behaviours and Histories of Agriculture, Food, and Nutrition, part of UK
Research and Innovation's Global Challenges Research Fund.





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



**Figure 1.** Map of Zambia, the black rectangle shows the study area. Map produced *de novo* by the authors



**Figure 2.** Generalized boundaries of the physiographic units from the 1947 soil vegetation map within the study area. Map produced *de novo* by the authors





**Figure 3.** Ila-Tonga Traverses of 1932 and the boundaries of the Sala Reserve. Note that the dotted lines join way-points with known locations, so generalize the actual route. Map produced *de novo* by the authors







**Figure 4.** Road Traverses of 1933 in Southern and Central provinces. Note that the dotted lines join way-points with known locations, so generalize the actual route. Map produced *de novo* by the authors

Road Traverses, Southern and Central Provinces, 1933





**Figure 5.** Road Traverses of 1934 in Southern and Central provinces. Note that the dotted lines join way-points with known locations, so generalize the actual route. Map produced *de novo* by the authors







**Figure 6.** Tree species recorded at cultivated sites (Sala Reserve excluded)

Dominant tree species at cultivated sites (not Sala reserve)



**Figure 7.** Numbers of references to particular species or associations of species as indicators in soil selection for cultivation. Light green symbols show cases where the selection is not first preference



**Table 1.** Trapnell's Stages of Agricultural Development in the Pemba district (Trapnell, 1934a).

| Stage | Management | | |
|---|---|---|---|
| 1 | Original dense scrub cultivation[1] | | |
| | Dense scrub cut and burned | | |
| | Yr 1. fm grown in ash, annual small extensions like this. gn/gb may be grown in small unburned patches | | |
| | Yr 2 – 4. mz and sg interplanted with cp | | |
| 2 | (a) Open bush fringe cultivation[2] | (b) Dambo cultivation | |
| | fm in small patches only with dense bush | sp/legumes in separate dambo margin gardens | |
| | land elsewhere broken with gn/gb | sg/mz in dambo grass | |
| | Separate sp gardens | sg preferred in clay soil at centre | |
| | Larger extensions than in (1) | 2 – 4 ac for 4 – 5 years | |
| | 4 – 8 under mz some sg for 2 – 3 yr | can exceed 7 yrs. | |
| 3 | Extensive open bush cultivation | | |
| | Larger areas of open bush are cleared, including poor and stony ground | | |
| | Some land is broken with gn, most directly to mz | | |
| | Small gn gardens are ploughed on neighbouring sites, abandoned after one year only | | |
| | No sg or fm grown, if sp planted, not in traditional mounds | | |
| | At the time of observation englargement was in process, more advanced clearances had stopped at about 25 ac. | | |
| | Manuring undertaken with the objective of cultivating permanently | | |

Crops are denoted as fm (finger millet), mz (maize), sorghum (sg), cp (cow pea), sp (sweet potato), gn (groundnut), gb (groundbean) [1] The labour requirement to clear scrub means that plots are limited to 2 – 4 acres. Over time return to land under secondary vegetation results in expansion of cleared land and transition to Stage 2. [2] There is a tendency to move to dambo cultivation to increase available land and reduce the labour demand for removing tree stumps.





**Table 2.** Development of main soil units used by Trapnell in published accounts. Within any one column the units have sometimes been reordered to show relationships between the legends, but the numbers show the original ordering in each source. Black fill in a row is where a new unit emerges with no corresponding unit in the Zanzibar abstract.

| Zanzibar abstract (Milne, 1935) | North Western Report (Trapnell and Clothier, 1937) | North Eastern Report (Trapnell, 1943) | Vegetation–Soil Map (Trapnell et al., 1947) |
|---|---|---|---|
| 1. Karroo Valley Group | 4. Lower Valley Soils | 5. Lower Valley Soils | 6. Lower Valley Soils |
| 2. Kalahari Group | 2. Kalahari Sands | NA[1] | 4. Kalahari Sands |
| 3. Plateau Group | 1. Plateau Soils | 2. Plateau Soils | 3. Plateau Soils |
|  |  | 1. Red Earths[2] | Red Earths and related red loam soils |
| 4. Upper Valley Group | 3. Upper Valley Soils | 4. Upper Valley Soils | 5. Upper Valley Soils |
|  | 5. Grey and Black Soils | 6. Dambo and Swamp soils | 7. Grey and Black Swamp Soils |
|  |  | 3. Lake Basin Soils | 2. Lake Basin Soils |
|  |  |  | 8. Escarpment Hill soils[3] |

[1] Kalahari Sands are restricted to the west of Zambia, and so the unit was not used in this part of the ecological survey. [2] Introduced as a variant of the Plateau soils. [3] There is a reference to these by Trapnell and Clothier (1937), but they are not treated as a distinct unit although they do appear implicitly in one of the map legend units ('*Isoberlinia globiflora—Brachystegia* woodlands of escarpment hill country passing into Lower Valley types', the third of the 'Southern Plateau types').