# Peer review of "Trapnell's Upper Valley Soils of Zambia: the production of an integrated understanding of geomorphology, ecology and land use"

_EGUsphere, 2024_

## Referee Comment (RC1)

**Review by Doyle McKey** (emeritus professor, University of Montpellier, France)

**General comments :** The manuscript analyzes the contribution of the Ecological Survey of Northern Rhodesia, carried out by Colin Trapnell and collaborators beginning in the 1930's. They focus on how his field observations led him to develop an integrated view of geomorphology, soil, plant communities, and agricultural potential. His work is still widely cited, and has been influential in developing our understanding of soils, vegetation, and agriculture in a part of Africa where soil fertility is often low (Bell 1982), but varies with geomorphology, with nutrient-poorer soils on uplands and nutrient-richer soils in bottomlands (Scholes 1990).

The manuscript argues that Trapnell's approach in fieldwork gave him particular insight into how soil conditions constrained agriculture, and on the adaptive value of traditional agricultural practices. The authors show that he analyzed not only how practices changed when farmers encountered environmental change, but also how they changed in response to social or economic change. They also show that he acknowledged the agency of African farmers, who made conscious decisions about which crops to grow, where, and when. His approach was thus more nuanced than that of what has been called « the ecological theory of development », a rather reductionist approach that some authors have considered to have been exemplified by his research. While the history of ideas about economic development, colonial legacies, and similar themes is rather far from my field of competence, these seem to be useful points. The detailed analysis and frequent citations of the survey's traverse records make for sometimes laborious reading, but they do help make the central points of the manuscript.

**Specific comments :** The authors argue that « close attention to Trapnell's experience could inform modern efforts to understand indigenous knowledge of African soils and their agricultural potential. » Modern work on soils and agricultural potential *has* been inspired and informed by Trapnell's work, and it would be useful to give some examples. There are many examples that can be cited. The examples I know are mostly from northwestern Zambia, where I have some field experience (I have no field experience in southern Zambia). The papers I know (e.g., Grogan et al. 2013 ; Mielke & Mielke 1982 ; Stromgaard 1984, 1985, 1986, 1989) thus cite Trapnell's later work in the northern part of present-day Zambia, from 1953 onwards (e.g., Trapnell 1953, 1959 ; Trapnell & Clothier 1957 ; Trapnell et al. 1976), rather than the early work in the survey in southern Zambia that is the subject of this paper. These papers analyze *chitemene* systems, described by Trapnell in 1953, and the *fundikila* and other mound-cultivation systems that seem to have been derived from *chitemene*. If the authors know of similar examples of modern work on soils and agricultural potential that were informed by the early work in the survey, in southern Zambia, these could be usefully cited and discussed.

References cited in this review :

Bell, R.H.V. (1982) The effect of soil nutrient availability on com- munity structure in African ecosystems. Ecology of tropical savannas (ed. by B J. Huntley and B.H. Walker), pp. 193-216. Ecological Studies; Springer, New York.

Mielke, H.W. and Mielke Jr, P.W., 1982. Termite mounds and chitemene agriculture: a statistical analysis of their association in southwestern Tanzania. *Journal of Biogeography*, 9(6) : 499-504.

Scholes, R.J., 1990. The influence of soil fertility on the ecology of southern African dry savannas. *Journal of Biogeography*, 17(4/5) : 415-419.

Stromgaard, P., 1984. The immediate effect of burning and ash-fertilization. *Plant and Soil*, 80 ; 307-320.

Stromgaard, P., 1985. A subsistence society under pressure: The Bemba of northern Zambia. *Africa*, *55*(1) : 39-59.

Stromgaard, P., 1986. Early secondary succession on abandoned shifting cultivator's plots in the miombo of South Central Africa. *Biotropica*, 18(2) : 97-106.

Stromgaard, P., 1989. Adaptive strategies in the breakdown of shifting cultivation: the case of Mambwe, Lamha, and Lala of northern Zambia. *Human Ecology*, 17 *:* 427-444.

Trapnell, C.G., 1953. The soils, vegetation and agriculture of north-eastern Rhodesia, Lusaka, Government Printer.

Trapnell, C.G., 1959. Ecological results of woodland burning experiments in Northern Rhodesia. *Journal of Ecology* 47: 129-168.

Trapnell, C. G., and Clothier, J. N. (1957). The Soils, Vegetation and Agricultural Systems of North Western Rhodesia. Report of the Ecological Survey. Government Printer, Lusaka, Northern Rhodesia.

Trapnell, M.T. Friend, G.T. Chamberlain, & H.F. Birch. 1976. The effects of fire and termites on a Zambian woodland soil. *Journal of Ecology* 64: 577-588.

Trapnell, C.G. and Clothier, J.N., 1937. The soils, vegetation and agricultural systems of North Western Rhodesia. Report of the ecological survey. *The soils, vegetation and agricultural systems of North Western Rhodesia. Report of the ecological survey.* Cited 201 times (10 since 2020, 67 since 2000)

**Technical corrections :**

The text is in general very clearly written, but I have made a (large) number of (small) corrections. Line-by-line corrections are listed below. The corrections and remarks concern several different points.

Many of these concern the use of commas. For instance, in many places there was a comma before a parenthesis. If a punctuation mark is associated with a parenthetical statement, it should come after the second parenthesis, never before the first parenthesis. In some sentences, a comma was needed and I suggested inserting a comma. In other places, I suggested deleting a comma. Presence or absence of a comma sometimes changes the meaning of a sentence. For example, when two bears are sitting at a table with a human named Bob, and one says « Let's eat, Bob ! », that is a friendly remark. But when the bear says « Let's eat Bob ! », that is cause for concern. In the manuscript, I saw no places where presence or absence of a comma dramatically changed the meaning of a sentence, but precision in the use of commas makes for easier reading.

There are a few places where a word (or more) was missing. I noted these places and either suggested a correction or asked the authors to clarify. There were also a few places where a pronoun was used, but it was unclear to what noun it referred.

In a few places where a finding of the decades-old report was presented, the authors use the present tense. In these cases, past tense or past-perfect tense should be used. I noted these places and suggested corrections or asked the authors to clarify.

There were a few minor grammatical errors (e.g., disagreement in number between subject and verb) and misspellings. I corrected these.

Finally, one kind of « correction » I made is more a stylistic suggestion than a correction : the use of « which » and « that ». In American usage, The difference between « which » and « that » depends on

whether the clause is restrictive or nonrestrictive : « In a restrictive clause, use 'that' ; In a nonrestrictive clause, use 'which'. ([https://www.grammarly.com/blog/which-vs-that/#:~:text=Which%20vs.%20that%3A%20What's%20the,a%20nonrestrictive%20clause%2C%20use%20which](https://www.grammarly.com/blog/which-vs-that/#:~:text=Which%20vs.%20that%3A%20What's%20the,a%20nonrestrictive%20clause%2C%20use%20which). –see that site for the difference between these two types of clauses. However, in British usage, the two words appear to be used more interchangeably ([https://www.reddit.com/r/grammar/comments/hwjlrp/that_vs_whicha_difference_in_british_english/](https://www.reddit.com/r/grammar/comments/hwjlrp/that_vs_whicha_difference_in_british_english/)).

Personally, I find the American usage more precise, and it seems to be catching on internationally.

Line 17 : remove the comma after « 1930s » (no punctuation just before a parenthesis)

Line 26 : WHAT « is focused on the East African Soil Map »--is the 'side-note' focused on this map, or is the 'account of pedology' focused on this map? I think you mean the latter. If so, the meaning would be clearer if you inserted a comma between « Africa » and « which ».

Line 80 : « was takes place » : Do you mean « was taking place » ? « took place » ? Or something else.

Line 83 : delete the comma after « scenery » (no punctuation just before a parenthesis)

Line 94 : « the first author was a missionary » : change to : « the first author of which was a missionary »

Line 95 : delete the comma after « work » (no punctuation just before a parenthesis)

Line 95 : insert a comma between « 4.2) » and « participated »

Line 110 : « has not been achieved, » : change to « had not been achieved, »

Line 111 : insert a comma between « officer » and « attributed »

Line 115 : insert a comma between « Moore » and « whose »

Line 117 : « In this context » : change to : « It was in this context »

Line 126 : delete the comma betwen « production » and « only »

Line 157 : delete the comma after « transcribed »

Line 158 : delete the comma after « assistance »

Line 272 : « Recall (section 3), that » : delete the comma

Line 273 : « has been an important factor » : change to : « had been an important factor »

Line 283 : delete the comma after « minute » (no punctuation just before a parenthesis)

Line 331 : « and landuse practices, » : change to : « and land-use practices, »

Line 337 : « wide ranging conversations » : change to : « wide-ranging conversations »

Line 381 : « field', (Clarke, 1936) » : delete the comma after « field » (no punctuation just before a parenthesis)

Line 399 : « mother, (Jaspan, 2017). » : delete the comma after « mother » (no punctuation just before a parenthesis)

Line 402 : « 2017) shifts » : insert a comma after the parenthesis

Line 408 : « 566 – 567) where » : insert a comma after the parenthesis

Line 408 : insert a comma between « undertaken » and « it »

Line 414 : « with one or more interpreter. » : change « interpreter » to « interpreters »

Line 424 : insert a comma between « However » and « we »

Line 424 : « individuals and the education or experience » change to : « individuals or the education or experience »

Line 425 : « which equipped them » : change to : « that equipped them »

Line 428 : « This is Cikabanga, » : Unclear. Is 'Cikabanga' the name of the language in which the term '*meninge skellem mouti*' appears ?

Line 456 : insert a comma between « mines » and « lies »

Line 469 : « in so far as this provide » : change « provide » to « provides »

Line 473 : « The information which Trapnell collected » : change « which » to « that »

Line 477 : « *B. flagristipulata* » : Spell out the name of the genus in full, as this species has not been mentioned previously in the manuscript.

Line 481 : « *Afrormosia spp*, » : First, « spp. » should not be in italics. Second, in the cited publication, was it « spp. » (plural, indicating more than one species) or « sp. » (singular, indicating one species)? It should be singular, as there is only one species. Third, « sp » should be followed by a period, not a comma. Fourth, while « *Afrormosia* » was used in the cited publication, it would be useful to add the current name, as follows : « (e.g. *Afrormosia spp*, [current recognized name *Pericopsis elata*, Fabaceae] and one … »

Line 482 : « *Setaria sp,* » : First, « sp » should not be in italics. Second, « sp » should be followed first by a period, then by a comma : « *Setaria* sp., »

Line 486 : « shifts, and rotations and » : change to : « shifts, rotations and »

Line 488 : « wild species which were eaten, » : change to : « wild species that were eaten, »

Line 500 : « *I. paniculata* » : Spell out the name of the genus in full, as this species has not been mentioned previously in the manuscript.

Line 512 : « *Compbretum* » : change to : « *Combretum* »

Line 520 : « The soil varies » : change to : « The soil varied » or to « The soil was observed to vary », because you are commenting on a report written in the past.

Line 533 : « Trapnell's observations on the around Kafushi » : a word appears to be missing. Do you mean « Trapnell's observations on the area around Kafushi » ?

Lines 543-544 : Spell out the generic name of these two species (or only the first one if they belong to the same genus). Also, put « B. hockii » in italics.

Line 546 : « gravel which is charateristic » : change « which » to « that » and change « charateristic » to « characteristic »

Line 548 : spell out the generic name of « *H. filipendula* »

Line 557 : « four year fallow » : change to « four-year fallow »

Line 557 : insert a comma between « site » and « which ». Also : « the community returned to the site, which was then abandoned. » : They returned for HOW LONG before the site was then abandoned?

Line 571 : « Nontheless » : change to : « Nonetheless »

Line 580 : « and others » : change to : « and at others »
Line 581 : « 1934) the demise » : insert a comma just after the parenthesis

Line 584 : It should be pointed out that when Trapnell refers to « anthills », these are in most (or all) cases termite mounds.

Line 588 : « technological change which was taking place » : change « which » to « that »

Line 590 : « but it was also associated » : What does « it » refer to here ? « the use of ploughing », or something else ?

Line 592 : « The traverse record, however, provide » : disagreement in number between subject and verb. Change to either : « The traverse records, however, provide » OR to : « The traverse record, however, provides »

Line 600 : « which notes » : change to : « that notes »

Line 643 : « contingencies which cause » : change to : « contingencies that cause »

Line 645 : « Western Report, Trapnell and Clothier (1937), he noted » : change to : « Western Report (Trapnell and Clothier, 1937), he noted »

Lines 655-656 : « Siegel (1989) shows that, for some, this was a deliberate ideological choice, the 'African Watchtower Movement', based among the Lamba, actively rejected urban life as a colonial innovation. » : This should be two sentences. E.g., "... Siegel (1989) shows that, for some, this was a deliberate ideological choice. The "African Watchtower Movement", based among the Lamba, actively..."

Line 684 : « concerns, which might » : change « which » to « that »

Lines 693-694 : « context which is entirely consistent » : change « which » to « that »

Lines 698-699 : « *Aformosia* » : correct to « *Afrormosia* » (current recognized name is *Pericopsis elata*)

Line 710 : insert a comma between « *filipendula* » and « were »

Line 711 : « and vegetation and Sweet Dambo » : change to : « and vegetation. Sweet Dambo »

Line 712 : « depended on a range of » : change to : « depended on the range of »

Line 714 : insert a comma between « Similarly » and « Colluvial »

Line 718 : « access, otherwise maize » : change to : « access. Otherwise, maize »

Lines 779-780 : « In the account of the soil classes the recently published East African Soil Map (Milne, 1936) is treated as normative, the point » : change to : « In the account of the soil classes the (then) recently published East African Soil Map (Milne, 1936) is treated as normative : the point »

Lines 784-785 : « in a state of maturity, and those stability, » : ?? One or more words are missing in this phrase

Line 787 : « poorly-drained conditions which » : insert a comma between « conditions » and « which »

Line 789 : « than these of the Southern Plateau which » : change to : « than those of the Southern Plateau,  which »
Line 792 : insert a comma between « ironstone » and « which »

Line 796 : « both with climate expressed in colour variation » : why "both"? What are the two things included in "both"?  Also, explain the relation between climate and colour variation. Unclear.

Line 799 : « your soils » : Why « your » soils ? Meaning unclear.

Line 800 : « Theses soils are correlated with » : Do you mean : « These soils are compared with » ? « These soils are considered similar to » ?

Line 805 : « extending of » : Do you mean « extending from » ?

Line 808 : « As noted in section **??** Trapnell » : there is a problem with the symbol after the word « section ». Also, insert a comma between the number of the section and « Trapnell »

Line 817 : « their larger base saturation and content of phosphate and nitrogen » : change to : « their larger base saturation and higher contents of phosphate and nitrogen »

Line 821 : « Trapnell and Clothier (1937), did not » : delete the comma

Lines 821-822 : « did not attempt these as mapping units. » : change to : « did not attempt to display these as mapping units. »

Lines 824-825 : « The Thorn soils were mainly on colluvial sandy loam material, generally more coherent
than the Transitional soils » : Meaning unclear. "more coherent" in what respect? Are you referring to some trait of the soil (e.g., structure)? Are you referring to the cohesioin of soil? Or are you referring to the coherence of the classification (these soils formed a more coherent class, i.e., less variable)?

Line 836 : « of the plateau » : these words should not be in italics.

Lines 840-841 : « Dambo Heads and Sweet Dambo of the Plateau Bush where gardens were established where *A. campylacantha* was found along with tall *Hyparrhenia* grass. » : Do you mean « Do you mean "... where gardens were established and where *A. campylacantha*..."? OR do you mean "...where gardens were established in places where *A. campylacantha*..."?

In either case, you should insert a comma between « Bush » and « where ».

Line 846 : « topography (dense scrub » : The second parenthesis is missing.

Line 847 : « and the cultivation period extended beyond » : Do you mean « Do you mean "...and the cultivation period can be extended beyond..."?

Lines 850-851 : « Sometimes followed by maize with or without sorghum. » : This is not a complete sentence (no subject or verb)

Line 858 : « the complexity which they observed » : change « which » to « that »

Line 866 : « the wood which was burned » : change « which » to « that »

Line 867 : « anthill, and where » : delete the « and ». Also, see my earlier comment about « anthill ».

Line 868 : « we sown » : change to : « were sown »

Line 871 : « The Ecological Survey Reports, (Trapnell and Clothier, 1937; Trapnell, 1943) » : delete the comma after « Reports » (no punctuation just before a parenthesis)

Line 872 : « the traditional systems which had been » : change « which » to « that »

Line 875 : insert a comma between « systems » and « which »

Lines 875-877 : « In the case of the Upper Valley System, however, they observe that, in the vicinity of the rail line the main priority is remediation in the light of rapid change which has already occurred. » : Verb tenses are tricky when you are describing a report that was published a long time ago. I suggest changing this sentence as follows : « In the case of the Upper Valley System, however, they observed that, in the vicinity of the rail line the main priority was remediation in the light of rapid change that had already occurred. »

Line 879 : insert a comma between « composts » and « could »

Lines 886-887 : « Three primary factors are identified, to which soils owe their characteristics. These are climate (past and present), parent material and the age of the land surface and nature of the changes that have taken place in relief. » : Where does the second factor end (after "parent material" or after "age of the lans surface") and the third factor begin? I BELIEVE that the second factor ends after « age of the land surface ». If so, insert a comma after « surface ». Then change « and nature of the changes » to : « and the nature of the changes. »

Line 888 : « The latter is key » : « latter » is used when speaking of the second of TWO things. "last" is used when speaking of the last of three or more things, as here ("three primary factors"). Change « latter » to « last ».

Line 889 : « says Trapnell (1943) *cut* » : insert a comma after « (1943) »

Line 896 : « environments where intense evaporation and soil moisture deficit results in » : change « results » to « result », as the subject of this verb is plural (« evaporation » and « deficit »).

Line 897 : « red earth soils correlated with those of » : Do you mean « red earth soils that are similar to those of » OR « red earth soils that correspond to those of » ?

Lines 906-907 : « As in the previous report, Trapnell explains the use of a *regional physiographic basis* for classification on the paucity of soil analytical data. » : "explains...on": unclear. I suggest writing something like: "As in the previous report, Trapnell explains that a *regional physiographic basis* was used for classification owing to the paucity of soil analytical data."

Line 907 : « he states that, when » : delete the comma.

Line 913 : « The notion of climatic sequences cutting across physiographic differences, » : Above (line 889), the structure is the inverse: physiographic differences "*cut across the broad zonal arrangement of climatic soil types.*"

Line 917 : « those formed in the most wettest conditions » : delete « most »

Line 920 : « representing, respectively tropical and more temperate » : either insert a comma after « respectively » or delete the comma after « representing »

Lines 929-930 : « The earlier correlation with Non-calcareous Plains Soils » : « correlation » is not quite the right word. I suggest writing : « The earlier correspondence of these soils to the Non-calcareous Plains Soils »

Line 931 : « Trapnell et al. (1947) notes » : change « notes » to « note » (there was more than one author).

Line 944 : « variation, and » : delete the comma.

Line 947 : « the characteristics of which vary » : Ambiguous. I suppose that "which" refers to the parent material provided by the Kalahari Sand, but from the sentence structure it could also refer to the underlying bedrock. Please clarify.

Line 950 : insert a comma between « 1930) » and « were »

Line 965 : « with A.P.G.Michelmorewho » : change to : « with A.P.G.Michelmore, who »

Line 967 : « bush, (Michelmore, 1934). » : delete the comma after « bush ».

Line 971 : « field observations, (Milne, 1947), were made : delete the comma before and after the parentheses.

Lines 971-972 : « conceptual model in the field which allowed » : change « which » to « that »

Line 978 : « reports, meant » : delete the comma

Line 991 : « based on process understanding » : Unclear. Do you mean "based on the understanding of process" ?

Line 1001 : « it was not irredeemable, (Trapnell, 1934a) and » : move the comma to read as follows : « it was not irredeemable (Trapnell, 1934a), and »

Lines 1008-1009 : « such as the Lamba (**?**) who » : What should replace the question mark ?

Line 1010 : « this overlooks » : change to : « this view overlooks »

Line 1011 : « factors which may influence » : change « which » to « that »

Lines 1011-1012 : « This underlines » : change to : « This consideration underlines »

Line 1012 : « soil and landuse which » : change to : « soil and land use that »

Line 1014 : insert a comma between « linguists » and  « as »

Line 1020 : insert a comma between « conditiosn » and « certainly »

Line 1026 : « concerns which » : change « which » to « that »

Line 1027 : « offers » : change to « offer » (the subject of this verb is « strategies », plural)

---

## Community Comment (CC1)

**Comments on Namwanyi et al. Trapnell's Upper Valley soils of Zambia: the production of an integrated understanding of geomorphology, pedology, ecology and land use**

https://doi.org/10.5194/egusphere-2024-315

Namwanyi *et al.* (2024) review at length the information on the soil of Zambia from the 1920s to present day. They focus on the legacy of C.G. Trapnell (1907–2004) whose ecological surveys of the territory from 1932 to 1943 included a wealth of information, not only on the vegetation but also on the soil, traditional agricultural practice and their variation from place to place. The vast Central African plateaux are mantled by thick residues of weathered rock that have remained in place perhaps since Tertiary times. This ancient oligotrophic soil now lacks minerals that can weather further to release plant nutrients. At the plateau margins, the land has been bevelled by erosion as streams cut back from the Luangwa and Zambesi valleys and the down-warped Kafue Flats. This erosion on a geological time-scale, so-called normal erosion', removed the ancient residues to leave relatively rich soil. Trapnell recognized this contrast; he called the bevelled margins 'Upper Valley'. The rich soil bore savanna vegetation quite different from the miombo woodland of the plateaux. Trapnell recognized that it had agricultural potential far greater than that of the plateaux, and he reported this to the government of the day.

In due course the government acted; it commissioned a soil survey of one large area of Upper Valley in Zambia's Eastern Province, and it fell to me to do it. Like the authors of this paper, I realized that Trapnell's distinguishing the Upper Valley soils from those of the plateaux on physiographic grounds was fundamental to pedological understanding, land use and agricultural development. It ran counter to the prevailing view in Europe and North America that climate was the main factor determining soil variation. Trapnell was a pedological pioneer who merits much greater credit than he has received from soil scientists. His profound appreciation of the relation between the vegetation, soil and African farming systems is a lesson for soil scientists today. They in turn will find a great deal in traditional knowledge that they can apply to solve current problems. I therefore welcome this paper by Namwanyi and his co-authors.

R. Webster

Rothamsted Research, Harpenden AL5 2JQ, UK

E-mail: richard.webster@rothamsted.ac.uk

---

## Author Response (AR1)

**Response to reviewers' comments**

**1  General comments**

We are glad for the opportunity to respond to comments on this paper. We note the handling editor's observation that it is an unusual paper for SOIL. We understand that, which is why we took advice from the Editors before submission. There is a growing interest in the history of soil science, particularly in the colonial context. This was apparent at the recent Centennial Meeting of the International Union of Soil Sciences (Florence, May 2024). There were contributions at the meeting by historians as well as soil scientists, and much interest in this from researchers who want to use legacy soil information from various periods. We therefore think that the time is ripe for soil science journals to take on papers which have professional historians collaborating with soil scientists so that the work is sound from the perspective of both disciplines.

We were also pleased to note the positive comment on the paper by Professor Webster, a soil scientist from Rothamsted Research, who worked in Zambia after Trapnell's time there.

**2  Reviewer 1: Doyle McKey**

**General comments**

The reviewer summarises the key points of the paper clearly. We are glad that he notes our response to those historians who use Trapnell's work in Zambia to exemplify the 'ecological theory of development'. We acknowledge the value of those studies but (and see reviewer 2) think that Trapnell's work requires more nuanced consideration.

We understand the reviewer's point that the style of the paper is detailed, but this reflects the historical methodology of 'close reading' of sources, which we think is essential here. For example, it helps us to see both how Trapnell's work influenced the development of the East-African soil map (Section 5.2) and why Trapnell appears to have played this down (lines 306 – 312 of the revised paper). Given our general observations above on the need for transdisciplinary work along the history/soil science axis, we hope that our contribution will help with mutual understanding by being historically rigorous and pedologically informed.

**Specific comments**

We agree that some references to the subsequent use of Ecological Survey publications in Zambia is useful. We have not found any that refer specifically to the Upper Valley, but we refer to some general examples in the introduction to the revised paper (see lines 17 – 21 of the revised paper). We note, also, that the published field records of the survey have barely been cited at all (see lines 36 – 44 in the revised paper). This provides context to our argument that a cross-disciplinary study is required (see lines 34 – 36 of the revised paper) and we return to this in the conclusion (lines 1071 – 1075).

**Technical corrections**

The reviewer's careful attention here is much appreciated. We have examined all his suggestions, and in each case we have made a revision to punctuation or presentation for improved clarity. Most of these corrections do not require further comment, but we expand on some below. Note that we quote the line number from the original submission to identify each comment, and give the corresponding line in the revised paper when we indicate our response.

- **Line 26**. We have rewritten this for clarity, see line 29 of the revised paper.

- **Line 80**. Changed to 'takes place', see line 92 of the revised paper.

- **Line 428**. We shall clarify that *Cikabanga* is the language of the excerpt, see line 441 of the revised paper.

- **Line 477**. We shall ensure, here and elsewhere, that the full generic name (e.g. *Brachystegia*) is given in the first reference to any species (e.g. *Brachystegia flagristipulata*).

- **Line 481**. At line 334 – 336 of the revised paper we explain that we use the Botanical names in the original Ecological Survey reports. It seems cumbersome to refer to the modern equivalents in the text, which is why we present the synonyms in Table S10 of the supplement. Note that we have checked the table of synonymy in the published traverse records and the modern equivalent of *Afrormosia angolensis* is *Pericopsis angolensis*.

- **Line 456** Here we have retained 'which' as this is not a definitive clause.

- **Line 557**. We have rewritten this for clarity. See line 571 – 573 in the revised paper.

- **Line 584**. We clarify that 'anthills' generally means 'termite mounds'. See line 600 of the revised paper.

- **Line 784**. We have edited this to read ' ... in a state of maturity and stability ...'. See line 817 of the revised paper.

- **Lines 796** We have edited this for clarity. See line 828 – 830 of the revised paper.

- **Line 799**. We deleted 'your'. See line 831 of the revised paper.

- **Line 800**. The verb 'to correlate' is used in geological and soil survey when two or more soil map legends, or working legends used by field surveyors, are compared to identify units which can be regarded as corresponding to each other. For example, Table C.4 in the first edition of the Booker Tropical Soil Manual presents the correlation of Soil Taxonomy and FAO soil groups. We have therefore not changed this, and inserted some text to explain the use of 'correlation' in this sense. See lines 833 – 836 of the revised paper.

- **Line 808**. This is left from a previous version of the paper. We have edited to read 'Trapnell compares them...', deleting 'As noted in section..'. See line 844 of the revised paper.

- **Lines 824 – 825** Trapnell uses 'coherent' here to say that the soil material was cohesive. We have clarified this. See line 861 of the revised paper.

- **Line 847**. We have clarified this. See line 883 of the revised paper.

- **Line 850–851**. We have rewritten this. See line 887 of the revised paper.

- **Lines 886-887**. We have rewritten this. See line 922–923 of the revised paper.

- **Lines 897, 929**. See response to comment on line 800 of the original paper above. We have left this as it was.

- **Line 913**. We have rewritten this. See line 950 of the revised paper.

- **Line 947**. We have rewritten this. See line 983–985 of the revised paper.

- **Line 991**. We have rewritten this. See line 1029 of the revised paper.

- **Line 1008 – 1009**. This was a reference which was mistyped and so not compiled in LaTeX. It was to the article by Speek and to the Central-Western report, see line 1045–1046 of the revised paper.

**3   Reviewer 2: Paul Smith**

**General comments**

We are glad that the reviewer finds our work to be useful, and concurs with our final assessment of the contemporary relevance of the Ecological Survey. We are particularly pleased given his key role in making the traverse records available through his collaboration with Colin Trapnell.

**Specific comments**

1. *Ecological concept of development* The 'Ecological concept of development' is a later term from historical appraisal of colonial science, rather than one used at the time (even though the Ecological Survey was widely referred to by its title, for example in reports of the Department of Agriculture). We agree that, as presented by Speek, it does not fit well as a descriptor of Trapnell's approach or conclusions, and we argue for a more nuanced understanding (section 4.7). In this regard we do not think we are in disagreement with the reviewer.

   We have revised section 4.7, in particular see lines 705 – 709 of the revised paper, to emphasize that, while the concept of an 'ecological' approach to agricultural development would not have been widely understood at the time, the two reports of the Ecological Survey could reasonably be interpreted as advancing such a concept, unsuccessfully as it turned out. Nonetheless, we think that Speek's characterisation of an ecological approach is reductive and fails to capture the sophistication of Trapnell's understanding of the farming systems he observed and the factors which shaped their development.

2. *The Lamba* We agree with the reviewer that Trapnell's attitude to African agricultural systems was very respectful, this is our main point in section 4.7. This distinguished him from many in the Agricultural department at the time where there was clearly considerable disagreement over many issues, not least whether European cultivation in Zambia should be encouraged at all. However, there are comments which appear in the reports of the Ecological Survey which do not reflect this nuanced understanding, and these have been picked up by authors such as Speek. For example, in the North Western report (Paragraph 90, starting on page 24), Trapnell and Clothier refer to The Lamba-Kaonde group, comprising the Bulima, the Lamba,... Generally speaking these are backward tribes, lacking in crafts and primitive in diet..'.

We are grateful to the reviewer for the references to the Traverse Records from around Solwezi which recognize specific features of Lamba practice which Trapnell regarded as well-adapted, and which show him distinguishing between practices of communities within the Lamba ethnolinguistic group. We have used these in revision as outlined below.

We have contrasted the generalized comment by Trapnell and Clothier (1937) with the more nuanced observations in the field records, observing that Trapnell noted variations in practice *within* these groups, varying from village to village (see line 697 – 704 of the revised paper). We contrast this with the generalised statement about the Lamba in the report. We think that this strengthens our critique of Speek. Note that we also refer to similar critiques by Moore and Vaughan, and by Tilley (lines 628, 696).

---

## Referee Report (RR1)

The authors have responded satisfactorily to my suggestions concerning content, and have made all the minor corrections and stylistic adjustments I recommended in my technical comments. I have made a few additional corrections of minor errors of form that I either missed in the first review or that appeared in the new text that was added during revision.

I guess my comments concerning the use of a hyphen (or not) in "land use" were not clear. The term takes a hyphen when it is a compound adjective modifying a noun, for example, "land-use change" (or "land-use practices", as on line 344). It does NOT take a hyphen when "use" is a noun and "land" is an adjective modifying it. In the title, for example, the correct form is "land use". Other places where the hyphen needs to be deleted to give two separate words : lines 46, 189, 341, 424, 519, 695, 749, 787, 1049, 1050, 1067.

Line 39 : I just did a Google Scholar search and found 31 citations of Smith and Trapnell (2001). It might be interesting to consult some of the additional ones not found in the Web of Science search.

Line 44 : You might consider citing these two references if you found them useful.

Line 96 : insert a comma between "landscape" and "was"

Line 105 : It is conventional to give the page number of a reference from which a direct quote was taken.

Line 159 : delete the comma after "records" (no punctuation just before a parenthesis)

Line 170 : insert a space between "P." and "Smith"

Line 181 : insert a comma after "Volume 1"

Line 190 : change "were" to "was" (subject of this verb is "information", singular)

Line 246 : "soil type of soil properties" : should "of" be "or" (or "and" ?)

Line 304 : Unless the error was already present in the text that is cited, change "usefuol" to "useful"

Line 324 : insert a comma after "(RML) "

Line 365 : "then" should be "than"

Lines 376-379 : This sentence is run-on. I suggest putting a period after "names" on line 377, and starting a new sentence with "Whilst"

Line 393 : delete the comma after "time" (no punctuation just before a parenthesis)

Line 409 : Delete the comma just before "would"

Line 499 : "…542. Practices…" change to : "…542). Practices…" (second parenthesis is missing)

Line 535 : spelling error. Correct to "characteristic"

Line 550 : spell out the generic name *Brachystegia*, as this is the first time this species is mentioned in the text.

Line 592 : What does "these" refer to ? The logical antecedents in the preceding sentence ("information", or "recent change") are both singular.

Line 667 : change "they appears" to "they appear"

Line 669 : "as such which" : insert a comma to read "as such, which"

Lines 670-671 : change "low-regard" to "low regard"

Line 696 : "and is also highlighted" : change to : "and as also highlighted"

Line 705 : delete the comma just before the last word of the line

Line 706 : "trying to advancing" : change to : "trying to advance"

Line 708 : delete the comma after "reports"

Line 761 : "did not attend, a letter" : change to : "did not attend. A letter"

Line 776 : Why are there parentheses around the number of the table ?

Lines 889-891 : Why do the words "main" and "village" bear an initial capital letter on line 889, but not on lines 890-891 ?

Figure 1, caption : "Map of Zambia, the black rectangle" : replace the comma by a semi-colon

Figure 6 and Figure 7 : all the "sp" in the species names should be "sp.", as this is an abbreviation, and "sp." should not be in italics.

Supplement : spell out the generic name for *B. tamarinoides* the first time it is mentioned. I don't think this species was already mentioned in the main text. I also don't recall previously seeing *B. mimosifolia*.

Table S1 : Entry for 14th June 1932 : last entry on the page : three Latin names are not in italics

I have not proofread the supplementary information in detail.

I like the supplementary table giving botanical names used by Trapnell and current names.

In the fourth line of the caption to this table (S10), change "is references" to "is referenced". Also, in the table itself, the spelling of *Julbernardia* has not been corrected. (in the table, this name is missing the second "r")

I don't think you have any table where you give the scientific names of the crops. The vernacular names might be ok for most crops, but some might be ambiguous. For example, in Table 1 of the main text, I suppose that "groundnut" is *Arachis hypogaea* and that "groundbean" is *Vigna subterranea*, but I would prefer to be sure. A supplementary table with scientific names of the crops mentioned would be a useful addition.

---

## Author Response (AR2)

School of Biosciences
University of Nottingham
Sutton Bonington Campus
Sutton Bonington
LE12 5RD
Loughborough
U.K.

20[th] September 2024

Dear Editor,

**EGUSPHERE-2024-315. Trapnell's Upper Valley Soils of Zambia: the production of an integrated understanding of geomorphology, pedology, ecology and land use**

On behalf of my colleagues I am glad to return our revised version of the above paper. Our responses to Reviewer 2 who requested these edits is given below.

With thanks and kind regards

Murray Lark

**Review of revised version Namwanyi et al.**

The authors have responded satisfactorily to my suggestions concerning content, and have made all the minor corrections and stylistic adjustments I recommended in my technical comments. I have made a few additional corrections of minor errors of form that I either missed in the first review or that appeared in the new text that was added during revision.

I guess my comments concerning the use of a hyphen (or not) in "land use" were not clear. The term takes a hyphen when it is a compound adjective modifying a noun, for example, "land-use change" (or "land-use practices", as on line 344). It does NOT take a hyphen when "use" is a noun and "land" is an adjective modifying it. In the title, for example, the correct form is "land use". Other places where the hyphen needs to be deleted to give two separate words : lines 46, 189, 341, 424, 519, 695, 749, 787, 1049, 1050, 1067. These changes have been made. The only places where "land-use" is hyphentated are line 348 and 427 in the revised paper where it serves as a compound adjective.

Line 39 : I just did a Google Scholar search and found 31 citations of Smith and Trapnell (2001). It might be interesting to consult some of the additional ones not found in the Web of Science search. This section has been rewritten, referrring to both Web of Science and Google Scholar, but making the point that, to date, the records have been cited to provide evidence on a range of questions from ethnobotany to past biodiversity, but not subjected to systematic close reading to study the survey process itself (lines 39 – 47 in the revised paper). Some of the new citations were found on Google Scholar.

Line 44 : You might consider citing these two references if you found them useful. Speek (2014) and Bowman (2011) are cited at line 49 of the revised paper.

Line 96 : insert a comma between "landscape" and "was" Done, line 100 of the revised paper.

Line 105 : It is conventional to give the page number of a reference from which a direct quote was taken. Done, line 109 of the revised paper.

Line 159 : delete the comma after "records" (no punctuation just before a parenthesis) Done, line 163 of the revised paper.

Line 170 : insert a space between "P." and "Smith" Done, line 174 of the revised paper.

Line 181 : insert a comma after "Volume 1" Done, line 185 of the revised paper.

Line 190 : change "were" to "was" (subject of this verb is "information", singular) Done, line 194 of the revised paper.

Line 246 : "soil type of soil properties" : should "of" be "or" (or "and" ?) Changed to "or", line 250 of the revised paper.

Line 304 : Unless the error was already present in the text that is cited, change "usefuol" to "useful"  Done, line 308 of the revised paper.

Line 324 : insert a comma after "(RML) " Done, line 328 of the revised paper.

Line 365 : "then" should be "than" Done, line 369 of the revised paper.

Lines 376-379 : This sentence is run-on. I suggest putting a period after "names" on line 377, and starting a new sentence with "Whilst" Done, lines 380 – 383 of the revised paper.

Line 393 : delete the comma after "time" (no punctuation just before a parenthesis) Done, line 397 of the revised paper.

Line 409 : Delete the comma just before "would" Done, line 413 of the revised paper.

Line 499 : "…542. Practices…" change to : "…542). Practices…" (second parenthesis is missing) Done, line 503 of the revised paper.

Line 535 : spelling error. Correct to "characteristic" Done, line 539 of the revised paper.

Line 550 : spell out the generic name *Brachystegia*, as this is the first time this species is mentioned in the text. Done, line 554 of the revised paper.

Line 592 : What does "these" refer to ? The logical antecedents in the preceding sentence ("information", or "recent change") are both singular. The sentence beginning on line 596 of the revised paper has been revised to clarify this.

Line 667 : change "they appears" to "they appear" Done, line 671 of the revised paper.

Line 669 : "as such which" : insert a comma to read "as such, which" Done, line 673 of the revised paper.

Lines 670-671 : change "low-regard" to "low regard" Done, line 674 of the revised paper.

Line 696 : "and is also highlighted" : change to : "and as also highlighted" Done, line 700 of the revised paper.

Line 705 : delete the comma just before the last word of the line Done, line 709 of the revised paper.

Line 706 : "trying to advancing" : change to : "trying to advance" Done, line 710 of the revised paper.

Line 708 : delete the comma after "reports" Done, line 712 of the revised paper.

Line 761 : "did not attend, a letter" : change to : "did not attend. A letter" Done, line 765 of the revised paper.

Line 776 : Why are there parentheses around the number of the table ? Removed, line 780 of the revised paper.

Lines 889-891 : Why do the words "main" and "village" bear an initial capital letter on line 889, but not on lines 890-891 ? Capitals removed, paragraph beginning at line 892 of the revised paper.

Figure 1, caption : "Map of Zambia, the black rectangle" : replace the comma by a semi-colon Done.

Figure 6 and Figure 7 : all the "sp" in the species names should be "sp.", as this is an abbreviation, and "sp." should not be in italics. Done.

Supplement : spell out the generic name for *B. tamarinoides* the first time it is mentioned. I don't think this species was already mentioned in the main text. I also don't recall previously seeing *B. mimosifolia*. Done.

Table S1 : Entry for 14th June 1932 : last entry on the page : three Latin names are not in italics Done.

I have not proofread the supplementary information in detail.

I like the supplementary table giving botanical names used by Trapnell and current names.

In the fourth line of the caption to this table (S10), change "is references" to "is referenced". Also, in the table itself, the spelling of *Julbernardia* has not been corrected. (in the table, this name is missing the second "r") Done.

I don't think you have any table where you give the scientific names of the crops. The vernacular names might be ok for most crops, but some might be ambiguous. For example, in Table 1 of the main text, I suppose that "groundnut" is *Arachis hypogaea* and that "groundbean" is *Vigna subterranea*, but I would prefer to be sure. A supplementary table with scientific names of the crops mentioned would be a useful addition. Done.